**1 Intra-annual variability of the Western Mediterranean Oscillation (WeMO)**
**2 and occurrence of extreme torrential precipitation in Catalonia (NE Iberia)**

Joan Albert **Lopez-Bustins** (1), Laia **Arbiol-Roca** (1), Javier **Martin-Vide** (1),
Antoni **Barrera-Escoda** (2) and Marc **Prohom** (1, 2)
(1) Climatology Group, Department of Geography, University of Barcelona
(UB), Barcelona, Spain.
(2) Department of Climatology, Meteorological Service of Catalonia,
Barcelona, Spain.

**10 Abstract**

In previous studies the Western Mediterranean Oscillation index (WeMOi) at daily
resolution has proven to constitute an effective tool for analysing the occurrence of
episodes of torrential precipitation over eastern Spain. The Western Mediterranean
region is therefore a very sensitive area, since climate change can enhance these
weather extremes. In the present study we created a catalogue of the extreme
torrential episodes (≥200 mm in 24 hours) that took place in Catalonia (NE Iberia)
during the 1951-2016 study period (66 years). We computed daily WeMOi values
and constructed WeMOi calendars. Our principal result reveals the occurrence of
50 episodes (0.8 cases per year), mainly concentrated in the autumn. We
confirmed a threshold of WeMOi ≤-2 to define an extreme negative WeMO phase
at daily resolution. Most of the 50 episodes (60%) in the study area occurred on
days presenting an extreme negative WeMOi value. Specifically, the most negative
WeMOi values are detected in autumn, during the second 10-day period of October
(11th-20th), coinciding with the highest frequency of extreme torrential events. On
comparing the subperiods, we observed a statistically significant decrease in
WeMOi values in all months, particularly in late October, and in November and
December. No changes in the frequency of these extreme torrential episodes were
observed between both subperiods. In contrast, a displacement of the extreme
torrential episodes is detected from early to late autumn; this can be related to a
statistically significant warming of sea temperature.

**31 Keywords**

Mediterranean, sea temperature, teleconnection indices, torrential precipitation,
WeMO.

## 1. Introduction

The Mediterranean seasonal precipitation regime is characterised by rainy winters and dry summers, linked to the westerly atmospheric circulation in winter and to the subtropical anticyclone belt in summer. Nevertheless, in some regions of the Mediterranean basin, the seasonal precipitation regime differs from the typically Mediterranean one; for example, most of eastern Iberia (Spain) displays a seasonal precipitation maximum in autumn, and a secondary one in spring (De Luis *et al.*, 2010; González-Hidalgo *et al.*, 2011). This bimodal precipitation pattern is recorded in few regions of the world. It only occurs over approximately 7% of the global land surface and is commonly associated with locations within the tropics (Knoben *et al.*, 2019). This bimodal behaviour in eastern Spain is mainly due to the physical geographic complexity of the Iberian Peninsula, which comprises several mountain ranges, all of which present different slope orientations. Furthermore, the Mediterranean Sea is practically cut off from other water bodies, which favours a higher sea surface temperature (SST) than in the Atlantic at the same latitude, especially in summer and autumn (Pastor *et al.*, 2015). This contributes to the development of high vertical gradients of air temperature in some months over the Mediterranean basin (Estrela *et al.*, 2008; Pérez-Zanón *et al.*, 2018). These physical geographical factors give rise to a high concentration of daily precipitation in the Mediterranean basin, i.e. torrential precipitation events, above all in the Western Mediterranean (Beguería *et al.*, 2011; Cortesi *et al.*, 2012; Caloiero *et al.*, 2019); all this reveals the need for water management in Spain to be based upon precipitation variability rather than on the precipitation mean (Lopez-Bustins, 2018). Heavy precipitation in the Western Mediterranean is mainly centred in eastern Spain, the south of France and the region of Liguria (NW Italy) (Peñarrocha *et al.*, 2002). These torrential events can cause dangerous floods and can have serious social and economic consequences, even human casualties, in the Mediterranean regions, e.g. in eastern Spain (Olcina *et al.*, 2016; Kreibich *et al.*, 2017; Nakamura and Llasat, 2017; Martin-Vide and Llasat, 2018) and in southern Spain (Gil-Guirado *et al.*, 2019; Naranjo-Fernández *et al.*, 2020). Climatological studies on torrential precipitation frequency and intensity are therefore relevant with regard to improving emergency plans and mitigating flood damage. Extreme precipitation

is expected to increase with global warming as a result of a greater atmospheric
water content (Papalexiou and Montanari, 2019); for instance, extreme peak river
flows are predicted to increase in Southern Europe during the current century
(Alfieri *et al*., 2015), and the frequency of heavy precipitation events is projected
to be higher for the 2011-2050 period (Barrera-Escoda *et al*., 2014).
Previous studies have associated extreme daily precipitation events in Spain with
synoptic patterns (Martin-Vide *et al*., 2008; Peña *et al*., 2015); these studies have
addressed several different tropospheric levels (Romero *et al*., 1999; Merino *et
al*., 2016; Pérez-Zanón *et al*., 2018). Furthermore, many studies have also
statistically correlated several teleconnection indices (El Niño Southern
Oscillation, North Atlantic Oscillation, Arctic Oscillation, Mediterranean
Oscillation, Western Mediterranean Oscillation, etc.) with precipitation series for
the Iberian Peninsula at different timescales (Rodó *et al*., 1997; Rodríguez-
Puebla *et al*., 2001; Trigo *et al*., 2004; Lopez-Bustins *et al*., 2008; González-
Hidalgo *et al*., 2009; Ríos-Cornejo *et al*., 2015a; Merino *et al*., 2016). Among
these indices, the Western Mediterranean Oscillation (WeMO) was found to be
the index most statistically and significantly correlated with annual, monthly and
daily precipitation on the littoral fringe of eastern Spain (Martin-Vide and Lopez-
Bustins, 2006; González-Hidalgo *et al*., 2009). The daily timescale of the WeMO
index (WeMOi) could constitute a potential tool for analysing the frequency of
torrential events in some regions of the Western Mediterranean basin.
Most torrential events in the Mediterranean region present a cyclonic centre at
surface level (Jansà *et al*., 1996; Rigo and Llasat, 2003). These cyclonic centres,
which are mainly mesoscale lows, can contribute to the structure of low-level
flows and therefore to the creation or intensification of a low-level warm and wet
current that can feed and sustain convection in favourable environmental
conditions (Jansà and Genovés, 2000; Jansà *et al*., 2000). Furthermore, the
Mediterranean Sea moistens and warms the low level of the atmosphere.
Consequently, the southerly to easterly flow that prevails before and during
torrential events in the Western Mediterranean transports the air under
conditional instability toward the coasts, where convection is often triggered by
an interaction between the flow and the orography. Studies based upon
mesoscale modelling, such as the research conducted by Lebeaupin *et al*.
(2006), show that an increase (or a decrease) in SST by several degrees
intensifies (or weakens) convection. In addition, the presence of a cut-off low in
the upper troposphere might be playing a significant role in the occurrence of
heavy precipitation, creating a cyclonic circulation in the lower troposphere, thus
enabling Atlantic air to be carried over the Mediterranean Sea. This warm and
very wet air in the lower layers impinges on the coastal mountains ranges and
the forced ascent is sufficient to trigger potential instability. This meteorological
configuration is accounted for the negative phase of the WeMO, which defines a
synoptic pattern prone to producing torrential precipitation and floods on the
Eastern Iberian coast. Daily precipitation amounts over 200 mm are not unusual
in such cases, particularly in eastern Spain, where many catastrophic floods are
related to the presence of a cut-off low (Llasat, 2009). Thus, these catastrophic
floods in the Northwestern Mediterranean basin are generally of synoptic origin
and are defined by the negative phase of the WeMO and enhanced by certain
mesoscale factors (Gilabert and Llasat, 2018).
The present study provides an exhaustive inventory of the most intense daily
precipitation events in Catalonia (NE Iberia) over the last few decades (1951-
2016) in order to provide a better understanding of their temporal distribution.
Moreover, we will analyse changes in frequency according to subperiods, since
the Western Mediterranean basin constitutes a global warming hotspot, where a
decrease in mean annual precipitation is expected for the following decades,
particularly in summer, together with a potential rise in storm-related precipitation
and drought duration (Christensen *et al*., 2013; Barrera-Escoda *et al*., 2014;
Cramer *et al*., 2018; Greve *et al*., 2018). The main aim of our study involves
creating a catalogue of extreme torrential events in Catalonia in order to establish
a period of high potential torrentiality in the area analysed at daily resolution. Most
studies delimit the wet season of a region within one or several months (Kottek
*et al*., 2006), and do not employ a smaller timescale than the monthly one.
Consequently, the present research attempts to use a more accurate timescale
than the monthly one in order to determine the period with the highest
accumulation of heavy precipitation episodes according to fortnights and 10-day
periods. The intra-annual variability of the daily WeMOi values may help to
establish the period with the highest propensity for torrential events in Catalonia.

Additionally, we analyse SST in order to establish a sea-atmosphere interaction to explain WeMOi values and changes in the frequency of events. Seawater constitutes an energy store, i.e. recharge areas, which can influence water vapour content and can intensify precipitation episodes (Pastor *et al.*, 2018; Iizuka and Nakamura, 2019) by means of a sea-atmosphere moisture exchange. Furthermore, a significant release of latent heat occurs during atmospheric convection over a warm sea like the Mediterranean at the end of summer and the beginning of autumn (Pastor *et al.*, 2015).

In section 2, we describe the main orographic and pluviometric features of the study area. The data and methods followed to calculate daily WeMOi values and construct the WeMOi calendar are explained in section 3. In section 4, the results of the intra-annual variability of torrential episodes, WeMOi values and sea temperature trends are analysed and discussed. Finally, in section 4 we derive the conclusions.

## 2. Study area

Catalonia covers an area of 32,100 km$^2$ in northeast Spain; it is physically separated from France by the Pyrenees (Figure 1). Altitude ranges from 0 (littoral) to 3,100 (northwestern Pyrenees) m.a.s.l. The Coastal and Pre-Coastal ranges, with an altitude ranging from 500 to 1,700 m.a.s.l., present a SW-NE orientation. On the western border, the Central Depression is approximately 200-300 m.a.s.l., constituting the driest part of the study area (350 mm annual mean precipitation) (Figure 2a). The wettest part of Catalonia is located in the Pyrenees, with an annual mean precipitation over 1,200 mm. In general terms, southern Lleida and Barcelona, as well as almost the entire province of Tarragona, make up the dry part of Catalonia (<700 mm). The rainy part of Catalonia (≥700 mm) comprises the province of Girona and the northern halves of the provinces of Lleida and Barcelona.

Catalonia's complex orography, as well as the fact that it comes under the influence of the Atlantic Ocean and the Mediterranean Sea, endow it with a highly heterogeneous spatial distribution of seasonal precipitation regimes throughout the study area. Using 70 monthly precipitation series (1951-2016) homogenized and provided by the Meteorological Service of Catalonia (SMC, 2017), we ascertained that, of the total of 24 possible permutations between winter, spring, summer and

autumn as dominant and subdominant precipitation seasons, 7 of these are detected
in Catalonia (Figure 2b) (Martin-Vide and Raso-Nadal, 2008). A clear predominance
of autumn precipitation can be observed, followed by spring precipitation, especially
in the coastal zone. The driest season on the coast is summer; however, the driest
time of year inland is winter. Many areas of the Pyrenees, above all in the east,
exhibit their maxima in summer as a result of convective precipitation.

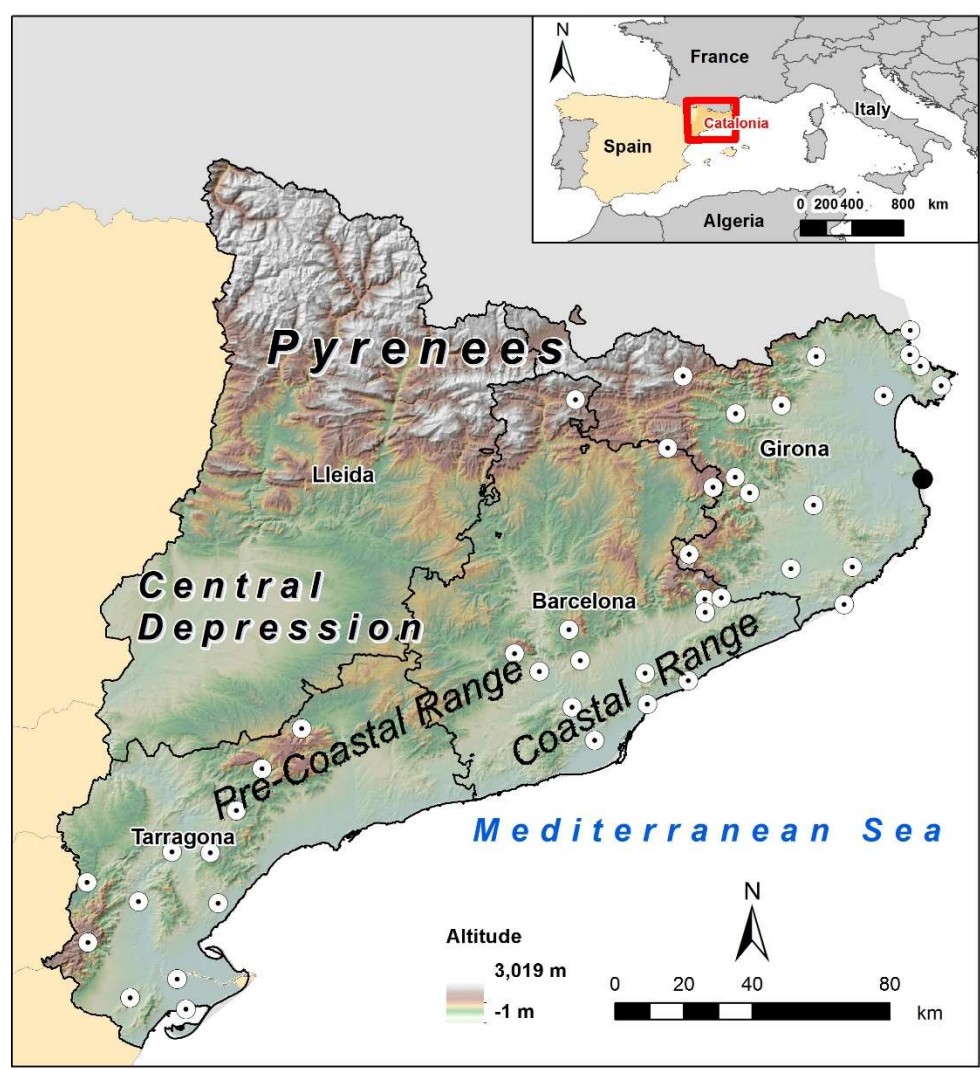


Figure 1. Location of Catalonia (NE Spain) within Europe, altitude and provinces.
The white dots indicate the 43 different weather stations that have recorded the
highest precipitation amount during an extreme torrential event at least once in
Catalonia during the 1951-2016 study period. The black dot indicates the location
of the sea temperature series. Base map provided by the Cartographic and
Geological Institute of Catalonia.

179                      (a)                                 (b)

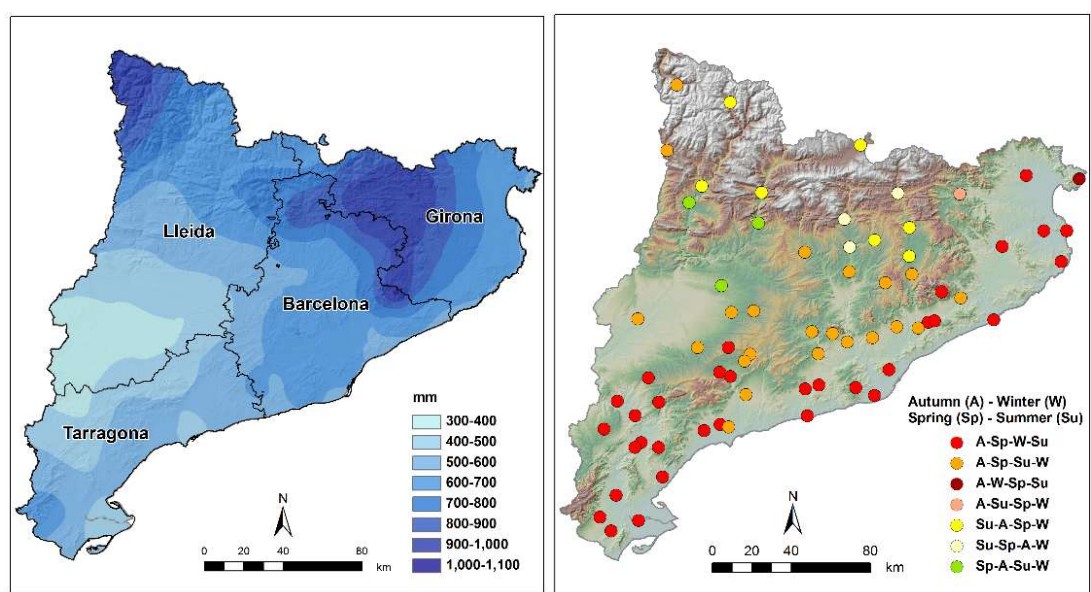


Figure 2. (a) Annual mean precipitation (mm) and (b) seasonal precipitation
regimes for 70 weather stations in Catalonia for the 1951-2016 study period.
Data source: SMC (2017). Base map provided by the Cartographic and
Geological Institute of Catalonia.

## 186   3. Data and methods

### 187   *3.1. Selection of torrential events*

Several studies have selected the torrential precipitation events in Spain based
on the threshold of 100 mm in 24 h (Pérez-Cueva, 1994; Martin-Vide and Llasat,
2000; Armengot, 2002; Riesco and Alcover, 2003; Martin-Vide *et al*., 2008).
Herein we chose the extreme torrential episodes (≥200 mm in 24 h) (Martin-Vide,
2002; Lopez-Bustins *et al*., 2016) that took place over Catalonia during the 1951-
2016 study period (66 years). We consider the threshold of 200 mm in 24 h to
present a natural risk in most cases, with significant consequences. Episodes
involving ≥100 mm in 24 h are more frequent, but sometimes have no direct
impact, or quite a negligible effect, because other factors are the main drivers of
floods, e.g. precipitation duration (Jang, 2015), initial soil moisture conditions and
hydrological parameters (Norbiato *et al*., 2008; Martina *et al*., 2009). Furthermore,
the area affected by episodes of ≥100 mm in 24 h is sometimes local and is

therefore not easily associated with advective synoptic patterns (Gilabert and Llasat, 2018).

In order to select the extreme torrential events, we considered all available precipitation data sources in Catalonia (Meteorological Service of Catalonia, Spanish National Meteorological Agency, Catalan Water Agency and Ebro Hydrographic Confederation). Thus, 1,466 weather stations were identified during 1951-2016, of which 986 were manually managed (67.3%) and provided one register per day, at 7 h UTC. Until 1987 the manual weather stations had constituted the only precipitation data source in Catalonia. The remaining 480 weather stations were automatic observatories, reporting hourly or semi-hourly data depending on the network and period. The 1988-2016 period was covered by both manual and automatic stations. We considered the pluviometric day as 7-7 UTC in both types of observatories in order to ensure a homogeneous criterion when selecting episodes along the whole study period and analysing any temporal changes in their frequency. We conducted an exhaustive spatial and temporal verification of the extreme torrential episodes identified. We tested the reliability of the events considering the daily precipitation recorded in neighbouring stations and examining the original handwritten observation cards. Furthermore, we rectified several episodes recorded by weather stations the day after the pluviometric day, and we eliminated events derived from the accumulation of precipitation for over one day.

The catalogue of extreme torrential events in Catalonia contains the following columns: date, maximum precipitation in 24 h, location, province and daily WeMOi value. Several observatories in Catalonia can occasionally register ≥ 200 mm in 24 h on one same date, but only the highest amount was taken into account. Finally, we obtained 50 extreme torrential events for consideration in the present study (Table 1). A total of 32 out of the 50 episodes (64%) have a decimal place of 0, and 10 out of the 50 episodes (20%) present a decimal place of 5. Most of these episodes were registered by manual weather stations prior to the 1990s. This is known as the rounding effect (Wergen *et al*., 2012): a weather observer rounds off the daily precipitation accumulation value during heavy precipitation events. This effect has no influence on the results of the present research.

| Date | Max RR (mm) | Location | Province | WeMOi value |
|---|---|---|---|---|
| **13 October 1986** | **430.0** | **Cadaqués** | **Girona** | **-2.22** |
| **11 April 2002** | **367.5** | **Darnius** | **Girona** | **-3.85** |
| 20 September 1971 | 308.0 | Esparreguera | Barcelona | -1.75 |
| 20 September 1972 | 307.0 | Sant Carles de la Ràpita | Tarragona | -1.58 |
| **09 October 1994** | **293.0** | **Cornudella de Montsant** | **Tarragona** | **-2.88** |
| 03 October 1987 | 291.0 | Castelló d'Empúries | Girona | -1.96 |
| **22 September 1971** | **285.0** | **Cadaqués** | **Girona** | **-2.19** |
| **19 October 1977** | **276.0** | **Cadaqués** | **Girona** | **-2.80** |
| **21 September 1971** | **275.0** | **Santa Maria de Palautordera** | **Barcelona** | **-2.21** |
| **18 October 1977** | **271.8** | **Camprodon** | **Girona** | **-2.21** |
| **21 October 2000** | **270.0** | **Falset** | **Tarragona** | **-2.26** |
| **07 November 1982** | **266.0** | **la Pobla de Lillet** | **Barcelona** | **-5.56** |
| 12 October 2016 | 257.0 | Vilassar de Mar | Barcelona | -1.86 |
| **05 March 2013** | **253.5** | **Darnius** | **Girona** | **-5.32** |
| **29 November 2014** | **253.5** | **Parc Natural dels Ports** | **Tarragona** | **-4.54** |
| **16 February 1982** | **251.2** | **Amer** | **Girona** | **-2.41** |
| 25 September 1962 | 250.0 | Martorelles | Barcelona | -1.52 |
| **04 November 1962** | **248.5** | **SantLlorenç del Munt** | **Barcelona** | **-2.79** |
| *02 September 1959* | *246.5* | *Cadaqués* | *Girona* | *-0.84* |
| **10 October 1994** | **245.0** | **Beuda** | **Girona** | **-2.33** |
| **22 October 2000** | **240.0** | **Tivissa** | **Tarragona** | **-2.50** |
| **12 November 1999** | **233.5** | **Castellfollit de la Roca** | **Girona** | **-3.00** |
| **06 January 1977** | **233.0** | **Girona** | **Girona** | **-2.22** |
| **20 December 2007** | **230.2** | **Parc Natural dels Ports** | **Tarragona** | **-3.54** |
| 06 October 1959 | 230.1 | Tossa de Mar | Girona | -1.36 |
| 03 October 1951 | 230.0 | Cornellà de Llobregat | Barcelona | -1.02 |
| 20 September 1959 | 230.0 | Gualba de Dalt | Barcelona | -1.49 |
| 11 October 1970 | 230.0 | Riudabella | Tarragona | -1.61 |
| **23 October 2000** | **229.0** | **Horta de Sant Joan** | **Tarragona** | **-2.41** |
| **26 September 1992** | **226.4** | **Amposta** | **Tarragona** | **-2.22** |
| **04 April 1969** | **226.0** | **Rupit** | **Barcelona** | **-2.21** |
| **12 November 1988** | **225.0** | **Corbera de Llobregat** | **Barcelona** | **-2.76** |
| 11 October 1962 | 223.0 | Sils | Girona | -1.20 |
| *20 November 1956* | *221.0* | *Cornellà de Llobregat* | *Barcelona* | *-0.45* |
| **06 November 1983** | **220.0** | **Terrassa** | **Barcelona** | **-2.34** |
| **19 October 1994** | **220.0** | **el Port de Llançà** | **Girona** | **-2.36** |
| *31 July 2002* | *218.2* | *Badalona* | *Barcelona* | *-0.13* |
| 13 September 1963 | 217.5 | l'Ametlla de Mar | Tarragona | -1.14 |
| *19 September 1971* | *217.0* | *Xerta* | *Tarragona* | *-0.97* |
| *17 September 2010* | *216.8* | *l'Ametlla de Mar* | *Tarragona* | *-0.60* |
| **17 October 2003** | **213.0** | **Vidrà** | **Girona** | **-2.48** |
| *09 June 2000* | *210.0* | *el Bruc* | *Barcelona* | *-0.23* |
| *31 August 1975* | *208.5* | *Santa Agnès de Solius* | *Girona* | *-0.15* |
| **29 January 1996** | **206.5** | **Fogars de Montclús** | **Barcelona** | **-2.37** |
| *09 October 1971* | *204.0* | *Miravet* | *Tarragona* | *-0.86* |
| **26 December 2008** | **202.5** | **Darnius** | **Girona** | **-2.84** |
| **07 May 2002** | **200.8** | **Godall** | **Tarragona** | **-2.47** |
| **07 October 1965** | **200.0** | **les Planes d'Hostoles** | **Girona** | **-2.12** |
| 27 October 1989 | 200.0 | el Port de la Selva | Girona | -1.90 |
| **01 November 1993** | **200.0** | **Portbou** | **Girona** | **-2.57** |

Table 1. Catalogue of extreme torrential events (≥200 mm in 24 h, 7-7 UTC) in
Catalonia (NE Iberia) during the 1951-2016 period. Max RR is the highest
precipitation accumulation of the episode. The events are classified according to
the extreme negative Western Mediterranean Oscillation (WeMO) phase (bold),
the negative WeMO phase and the slight negative WeMO phase (italics).

*3.2. Daily WeMOi values*

The WeMOi is a regional teleconnection index defined within the Western Mediterranean basin (Martin-Vide and Lopez-Bustins, 2006) and already used in a wider range of studies (Azorin-Molina and Lopez-Bustins, 2008; Vicente-Serrano *et al*., 2009; Caloiero *et al*., 2011; El Kenawy *et al*., 2012; Coll *et al*., 2014; Ríos-Cornejo *et al*., 2015b; Lana *et al*., 2017; Jghab *et al*., 2019). WeMOi values are computed by means of surface pressure data from the San Fernando (SW Spain) and Padua (NE Italy) weather stations (Figure 3); the synoptic window 30º-60ºN - 15ºW-20ºE is found to best represent WeMO phases (Arbiol-Roca *et al*., 2018). Pressure data for both series were extracted from Martin-Vide and Lopez-Bustins (2006), who performed a statistical treatment of homogenization and the Climatology Group (University of Barcelona) periodically update the data. The positive phase of the WeMO corresponds to the anticyclone over the Azores encompassing the southwest quadrant of the Iberian Peninsula and low pressures in the Gulf of Genoa (Figure 3a); its negative phase coincides with an anticyclone located over Central or Eastern Europe and a low-pressure centre, often cut off from the northern latitudes, within the framework of the Iberian southwest (Figure 3b). Martin-Vide and Lopez-Bustins (2006) found that the WeMOi was significantly and statistically correlated with precipitation over areas that were weakly influenced by the North Atlantic Oscillation (NAO): these areas are the northernmost and easternmost parts of Spain; precipitation over the Cantabrian fringe (northern Spain) is strongly and positively correlated with the WeMOi, and precipitation over the Spain's eastern façade is strongly and negatively correlated with the WeMOi.

271            (a)                       (b)

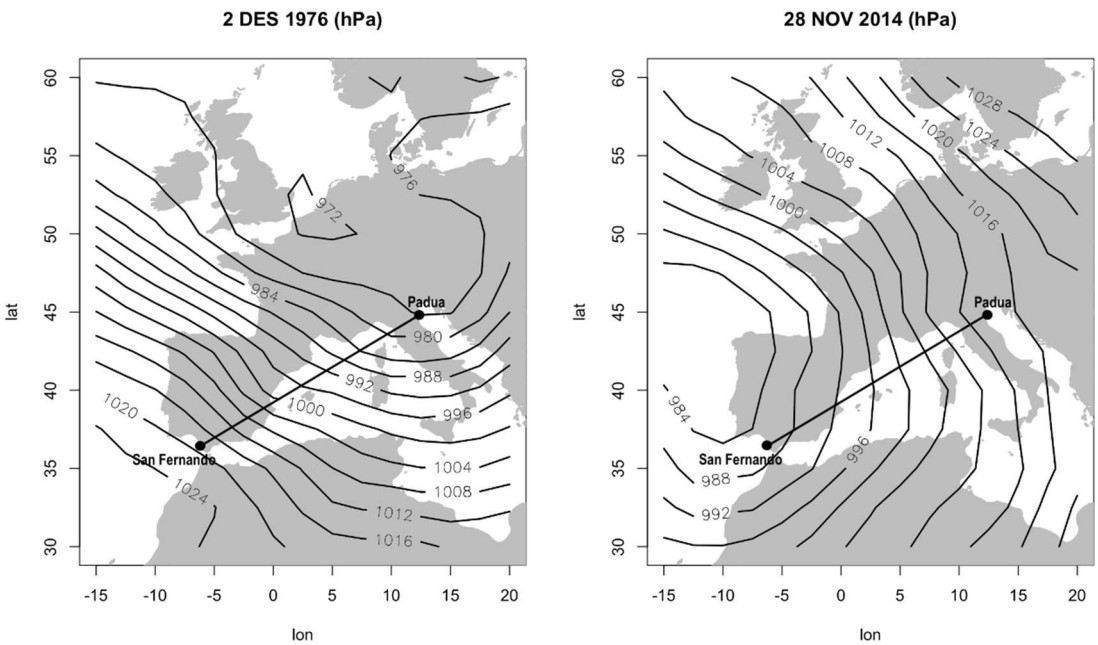


Figure 3. (a) Most extreme positive phase of the Western Mediterranean Oscillation
(WeMO) in a daily synoptic situation during the 1951-2016 study period (2nd
December 1976). (b) Most extreme negative WeMO phase in a daily synoptic situation
during the 1951-2016 study period (28th November 2014). Data source: NCEP
Reanalysis data provided by the NOAA/OAR/ESRL PSD, Boulder, Colorado, USA.
Application of the daily WeMOi is a methodological contribution by Martin-Vide
and Lopez-Bustins (2006). It converts the low-frequency feature of the
teleconnection patterns into a high-frequency mode. It is suitable for application
both to the regional scale of the WeMO teleconnection pattern and the lesser
variability of atmospheric pressure at Mediterranean latitudes. Patterns have
rarely been used at daily resolution (Baldwin and Dunkerton, 2001; Beniston and
Jungo, 2002; Azorin-Molina and Lopez-Bustins, 2008; Liu *et al*., 2018). The
method selected consists of previously standardizing each series of the dipole. It
is necessary to use the daily mean and standard deviation of the 1961-1990
reference period of all days of the year (January 1st 1961 – December 31st 1990).

289       For example, the WeMOi on January 1st 1981


$$Z\,WeMOi\;Jan\,1st\,1981 = \frac{P\,Jan\,1st\,1981\,SF - \overline{X}\,1961\_1990\,SF}{S\,1961\_1990\,SF} - \frac{P\,Jan\,1st\,1981\,PD - \overline{X}\,1961\_1990\,PD}{S\,1961\_1990\,PD},$$
where P is pressure, SF, San Fernando, PD, Padua, $\bar{X}$, mean, and $S$, standard
deviation.

This calculation method, which considers all days of the year in the reference
period, enables all Mediterranean flows (negative WeMO phase) to be detected,
even if they are very weak. Otherwise, these moderate Mediterranean winds
would not be detected in autumn, since the WeMOi means are clearly negative
during this season. Likewise, the weak Mediterranean flows would be
overestimated in winter due to the high WeMOi mean during the coldest months.
According to previous studies (Martin-Vide and Lopez-Bustins, 2006; Azorin-
Molina and Lopez-Bustins, 2008), in the histogram of daily WeMOi frequencies,
WeMOi values between -1.00 and 1.00 are considered to constitute a neutral
WeMO phase, values ranging from 1.00 to 1.99 are considered as a positive
WeMO phase, those between -1.99 and -1.00 as a negative WeMO phase,
values ≥2.00 are deemed to represent an extreme positive WeMO phase and
those ≤-2.00 to indicate an extreme negative WeMO phase. The most positive
WeMOi value (+5.99) of the 1951-2016 study period refers to December 2nd
1976 (Figure 3a), when an intense precipitation episode was recorded in the
Basque Country (northern Spain), according to ECA dataset (Klein Tank *et al.*,
2002; Cornes *et al.*, 2018). The most negative WeMOi value (-5.97) during the
1951-2016 period corresponds to November 28th 2014 (Figure 3b), when 253.5
mm was registered in the *Parc Natural dels Ports* (Tarragona) during the following
day (Table 1). Lana *et al.* (2016) studied the statistical complexity and
predictability of the WeMOi and demonstrated the Gaussian distribution of this
index. Most daily WeMOi values are negative (55%) and two thirds of the 23,996
days displaying WeMOi values correspond to a neutral WeMO phase (Figure 4).
The positive (negative) WeMO phase was detected in 16.5% (17.2%) of the total
days presenting a WeMOi value. The extreme WeMOi values, both positive
(5.2%) and negative (3.9%), represent less than 10% of the total number of days
for which WeMOi values are available. Daily NAO index (NAOi) values are also
used for comparison with WeMOi values and to enhance the role played by the
WeMO in torrential precipitation. Following the calculation method based on daily
WeMOi values, daily NAOi values are computed by means of surface pressure
data from the San Fernando (SW Spain) and Reykjavík (SW Iceland) weather
stations; the data for Reykjavík were provided by the ECA dataset (Klein Tank *et*
*al*., 2002). The NAOi values present the same percentage as that of the negative
WeMOi daily values (55.1%) and almost half of the days are around 0. The
distribution of the daily values of the NAOi presents more extreme positive and
negative values than the WeMOi distribution, 12.4 vs 5.2% and 8.7 vs 3.9%,
respectively (Figure 4).

**Daily values 1951-2016**


Figure 4. Frequency histogram of all daily WeMO index (WeMOi) values and
North Atlantic Oscillation index (NAOi) values during the 1951-2016 study period.
*3.3. Construction of calendars*
Construction of calendars is a common procedure in climatological studies (Soler
and Martin-Vide, 2002; Azorin-Molina and Lopez-Bustins, 2008; Meseguer-Ruiz
*et al*., 2018). They enable the intra-annual variability of the climate variable to be
visualised. We computed daily WeMOi values for the 1951-2016 (66 years) study
period, constructing two WeMOi calendars based upon the mean values obtained
for each month, a 15-day period (i.e. a fortnight) and a 10-day period; the latter
timescale corresponds approximately to the baroclinic prediction period (Holton,
2004). The first climate calendar will show the annual cycle of the WeMOi values
according to months (12 values), the second will display a more detailed intra-
annual oscillation with 24 values and, finally, the 36 WeMOi values derived from
the 10-day calendar will enable the slightest intra-annual variations in the WeMOi
to be detected. We will add to these calendars all the extreme torrential events in
order to observe correspondences between WeMOi values and heavy
precipitation events along the year. In order to detect any changes in the
calendars throughout the study period, we consider two subperiods for the
construction of two additional calendars: 1951-1983 (33 years) and 1984-2016
(33 years). We statistically tested the mean WeMOi values according to
subperiods in order to detect statistically significant differences. This statistical
significance is computed by means of a Normal distribution test according to
several confidence levels: 95.0% (Z=1.960), 99.0% (Z=2.576) and 99.9%
(Z=3.291).
Additionally, we analysed these calendars according to subperiods, together with
changes in SST and subsurface temperature at several depths (20, 50, and 80
m.b.s.l.) at a site located on the coast of Girona province (Figure 1). These data
constitute a reference series of sea temperature observations for Spain and for
the Mediterranean basin due to their long temporal range (almost half a century)
and to their availability at several subsurface levels (Salat *et al.*, 2019); the data
on the 1973-2017 period were provided by the Meteorological Service of
Catalonia. We calculated monthly temporal trends in sea temperatures using the
least-square linear fitting, and we estimated the statistical significance by means
of the Mann–Kendall non-parametric test (Sneyers, 1992). The standardized
values (Z) of sea temperatures were computed at 10-day resolution, and the Z
differences were obtained between two 5-yr subperiods from the beginning and
the end of the 1973-2017 period: 1973-1977 and 2013-2017; we showed the Z
differences for the months of the wet season (September, October and
November) for most of Catalonia (Figure 2b), and also for December in order to
detect a potential temporal shift of sea warming rates towards the early winter.
**4. Results and discussion**
*4.1. Frequency and temporal evolution of the extreme torrential events*
During the 1951-2016 period, 50 episodes presenting ≥200 mm in 24 h were
detected (0.8 cases per year) in Catalonia (Table 1); these were mainly
concentrated in the Eastern Pyrenees (Girona) and southern Catalonia
(Tarragona) (Figure 1), where mountain ranges run in a N-S direction, constituting
an orographic barrier to the humid easterly flows (Lopez-Bustins and Lemus-
Canovas, 2020). In the province of Lleida no maximum values for precipitation
episodes have been recorded, because this province is less influenced by
easterly flows as a result of its continental features. Other parts of Iberia register
a higher frequency of extreme torrential events, e.g. in the Valencia Region,
eastern Spain, there were 2 cases per year during the 1971-2000 period (Riesco
and Alcover, 2003). The highest frequency of torrential events (≥100 mm in 24 h)
over the Iberian Peninsula also corresponds to the Valencia Region, where more
than one case per year can be recorded by one same observatory (Pérez-Cueva,
1994) and approximately 11 cases per year by all the stations in the Valencia
Region (Riesco and Alcover, 2003). Catalonia exhibits a lower frequency of these
torrential events (i.e. ≥100 mm in 24 h), 5-6 cases per year for the whole region
(Martin-Vide and Llasat, 2000; Lopez-Bustins *et al*., 2016). The highest
precipitation amount during 7-7 UTC ever recorded in Catalonia is 430 mm. This
occurred in Cadaqués (Cape Creus, in the easternmost part of the Iberian
Peninsula) on October 13th 1986. It was an extraordinary episode which also
affected the region of Pyrénées-Orientales (S France) (Vigneau, 1987), albeit
with a lower amount of precipitation than that produced by other extreme torrential
events of over 800 mm in Liguria Region (NW Italy), Valencia Region (E Spain)
and this region of Pyrénées-Orientales (Peñarrocha *et al*., 2002).
Most of the episodes in Catalonia (60%) (30 events) took place in an extreme
negative (≤-2.00) WeMO phase (Figure 5), whereas less than 4% of the total
number of days with WeMOi data showed a value equal to or lower than -2.00
(Figure 4). Moreover, 24% (12 events) of the episodes occurred in a negative (-
2.00, -1.00] WeMO phase. The remaining 8 events (16%) took place in a slightly
negative (-1.00, 0.00) WeMO phase. No extreme torrential episodes presenting
a positive WeMOi value occurred in Catalonia during the study period.
Furthermore, Martin-Vide and Lopez-Bustins (2006) found no positive daily
WeMOi values for torrential episodes (≥100 m in 24 h) in Tortosa (south
Catalonia) during the 1951-2000 period. On the other hand, the maximum
concentration of extreme torrential events according to NAOi values falls within
the interval (-1.00, 0.00), and both negative and positive NAOi values can account
for an event. This result demonstrates the fact that daily WeMOi values are more
useful than daily NAOi values. This is further evidenced by the fact that only 24%
of the total number of events took place during an extreme negative (≤-2.00) NAO
phase, whereas this percentage rises to 60% in an extreme negative WeMO
phase.

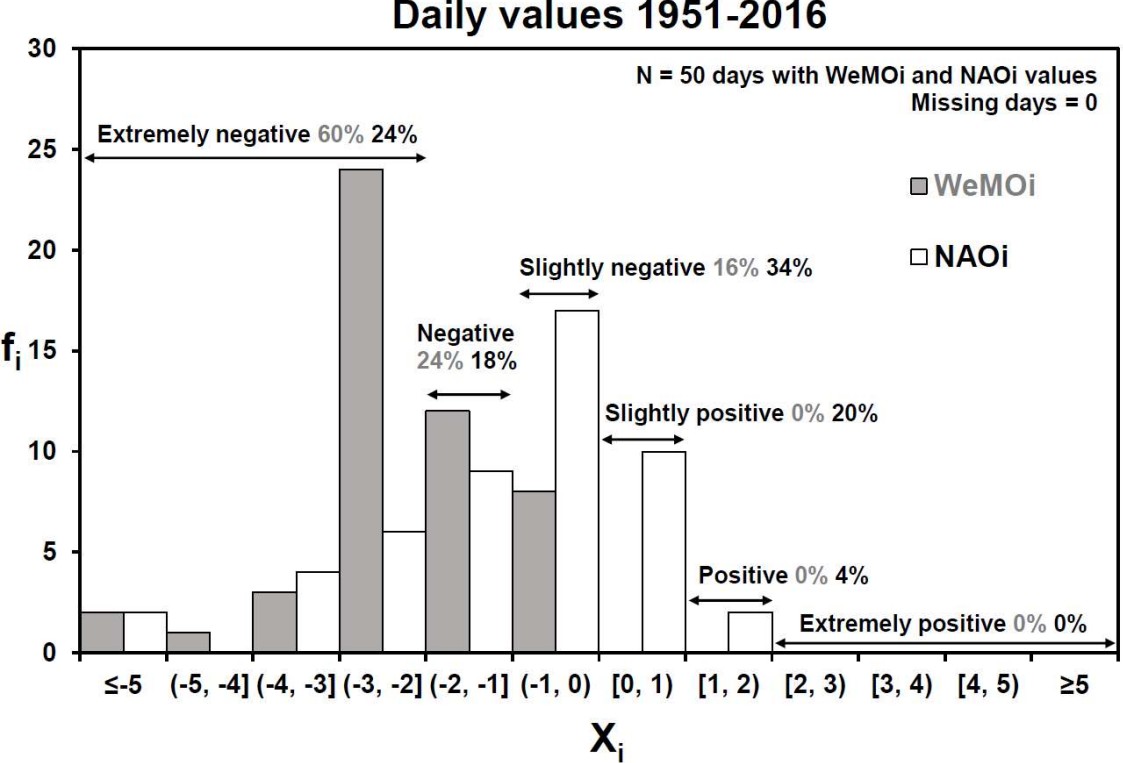


Figure 5. Frequency histogram of the daily WeMOi and NAOi values of the 50
extreme torrential events recorded in Catalonia during the 1951-2016 study period.
Most of the years in the 1951-2016 period present no episodes, or only one (Figure 6);
in six years there were 2 or 3 episodes, depending on the year, and in just two
years (1971 and 2000) we detected over 3 episodes in one year. The greatest
accumulation of cases can be observed in 1971, when a long-lasting torrential
episode exceeded the threshold of 200 mm in 24 h during four consecutive days
in September, with another one-day episode occurring in October. The former is
one of the most noteworthy episodes recorded in Catalonia (Llasat, 1990; Martin-
Vide and Llasat, 2000) in the last few decades. It started on September 19th in
southern Catalonia and ended on September 22nd in the northeast of the study
area (Llasat et al., 2007). During the last decade, there has been no more than
one episode in one single year. However, for torrential events (≥100 mm in 24 h)
in Catalonia, Lopez-Bustins et al. (2016) detected a 45% increase in cases
between the 1950-1981 and 1982-2013 subperiods. In accordance with this rise
in torrential precipitation events, many studies on Iberian precipitation are
showing an increase in precipitation of Mediterranean origin in eastern Spain
(Miró *et al*., 2009; Lopez-Bustins *et al*., 2008; De Luis *et al*., 2010); this
contributes to an increase in precipitation variability over the Western
Mediterranean (Hartmann *et al*., 2013, Caloiero *et al*., 2019). On the other hand,
non-statistical temporal trend is observed in the annual frequency of the extreme
torrential episodes (i.e. ≥200 m in 24 h) in Catalonia during the study period
(Figure 6). This is in line with Llasat *et al*. (2016), who found non-statistical
temporal trends in extreme daily precipitation in Catalonia.

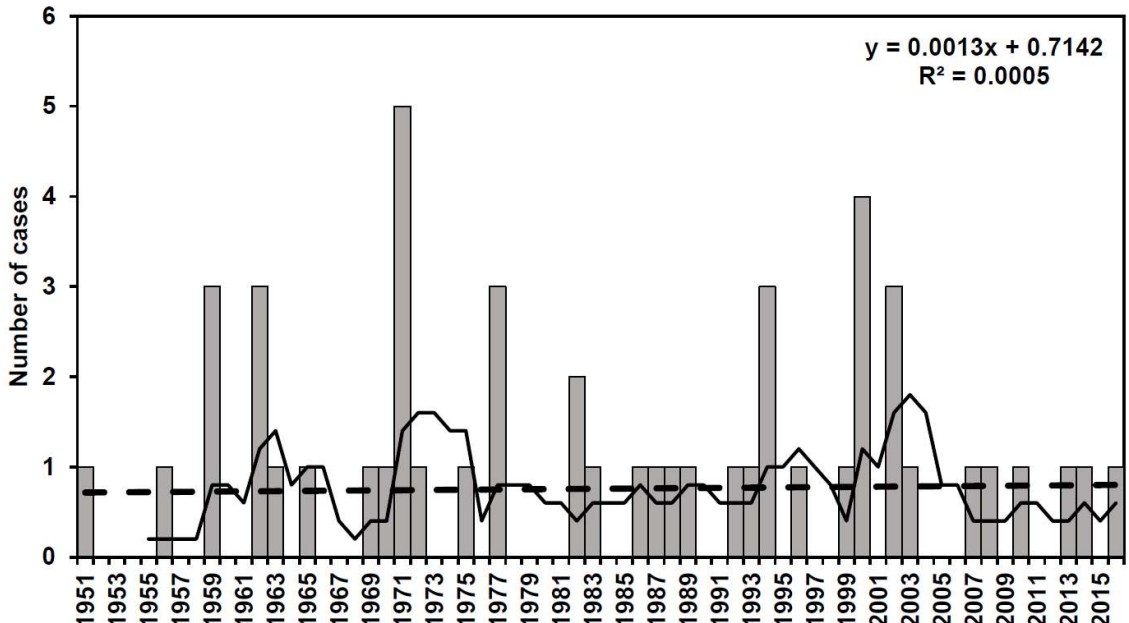


Figure 6. Temporal evolution of the annual frequency of extreme torrential events
(≥200 mm in 24 h) throughout the 1951-2016 study period. The figure shows the
linear regression (dashed line) and 5-yr running mean (black line).
*4.2. Calendars of the daily WeMOi values*
The lowest WeMOi values are detected in autumn, especially in October (-0.38)
(Figure 7a), usually with humid easterly flows from the Mediterranean Sea. This
explains why autumn and October are the wettest season and month,
respectively, on most of Spain's eastern façade (De Luis *et al*., 2010). The
greatest accumulation of extreme torrential events in Catalonia is in October, with
19 events (38% of all cases). This is coherent with subsurface sea temperature,
which reaches its annual maximum in autumn (not shown). September also
shows a remarkable accumulation of events (11 cases), displaying the second
lowest WeMOi monthly value (-0.29). Positive WeMOi values are observed from
December to March, with very few events occurring. Sea temperature decreases
after the wet season, and the first months of the year constitute the period when
sea waters are the coldest (not shown). Additionally, WeMOi values are very high
in January and February, and the precipitation-convection phenomenon can
therefore be halted by a strong decrease in SST (Lebeaupin *et al*., 2006).
Although negative WeMOi values are detected from April to November, very few
episodes are registered in late spring and summer; the predominance of
atmospheric stability during the warm season reduces the chances of extreme
torrential events occurring over the study area. At the fortnightly timescale, we
detected the minimum WeMOi value (-0.39) during the second half of October
(Figure 7b). The greatest accumulation of episodes, however, is in the first half
of October. The lowest WeMOi values are found from September 16th to October
31st. This short period of the year (46 days) accumulates over one half of the
total amount of extreme torrential events (28 cases, 56%). The most positive
WeMOi values are detected in the winter months, particularly from January 1st to
February 15th, and only 2 episodes are registered.
At the 10-day timescale, we observed the WeMOi minimum value (-0.45) from
October 11th to 20th (Figure 7c). This 10-day period also presents the largest
accumulation of extreme torrential events in Catalonia (8 cases; 16% of the total
number of cases). At least 4 cases are registered in each 10-day period from
September 11th to November 10th. This period of the year (61 days) accumulates
two thirds (33 cases, 66%) of all extreme torrential events. WeMOi values are
lower than -0.20 from August 1st to November 10th, fitting well with the period of
highest frequency of extreme torrential events in Catalonia. From August 1st to
September 10th, only 2 cases are registered due to the above-mentioned
atmospheric conditions in summer. From September 11th to November 10th,
favourable conditions can arise for the occurrence of extreme torrential events in
Catalonia: a high SST in the Western Mediterranean Sea and the early cut-off of
subpolar lows travelling to Mediterranean latitudes (Estrela *et al*., 2008; Lopez-
Bustins, *et al*., 2016; Pérez-Zanón *et al*., 2018). The positive WeMOi values are

observed from December to March and each 10-day period presents either no episode or only a single one. The most positive WeMOi value is observed from January 1st to 10th (+0.38); this indicates the total predominance of the positive phase of the teleconnection during these days, according to the 1951-2016 study period (Figure 8a). During this 10-day period, the occurrence of extreme torrential events in eastern Iberia is strongly inhibited by the NW atmospheric circulation over the study area; sea waters are cold and the Genoa low is well represented. The remaining 10-day periods in winter also present a predominance of the western circulation over the Iberian Peninsula. This pattern causes positive pressure differences between the Gulf of Cadiz (at a lower latitude) and the North of Italy (at a higher latitude), which produces positive WeMOi values and inhibits precipitation in eastern Iberia because of its location in the lee of the westerlies. On the other hand, the mean sea level pressure (SLP) map from October 11th – 20th shows a predominance of the negative WeMO phase, with humid easterly flows over Iberia, low pressure usually located in the Western Mediterranean basin, and a blocking anticyclone over Central and Eastern Europe (Figure 8b).

This is approximately 60% of the year falling under negative WeMOi values at monthly N= 8 (out of 12) (Figure 7d), fortnightly N = 14 (out of 24) (Figure 7e), and 10-day N = 23 (out of 36) (Figure 7f) timescales. The linear regression between negative WeMOi values and episodes is statistically significant at all timescales, providing an R of -0.73 (Figure 7d), -0.72 (Figure 7e) and -0.72 (Figure 7f). There is a statistically significant increase in the occurrence of events as the WeMOi value decreases. The linear fitting is especially significant at 10-day resolution.

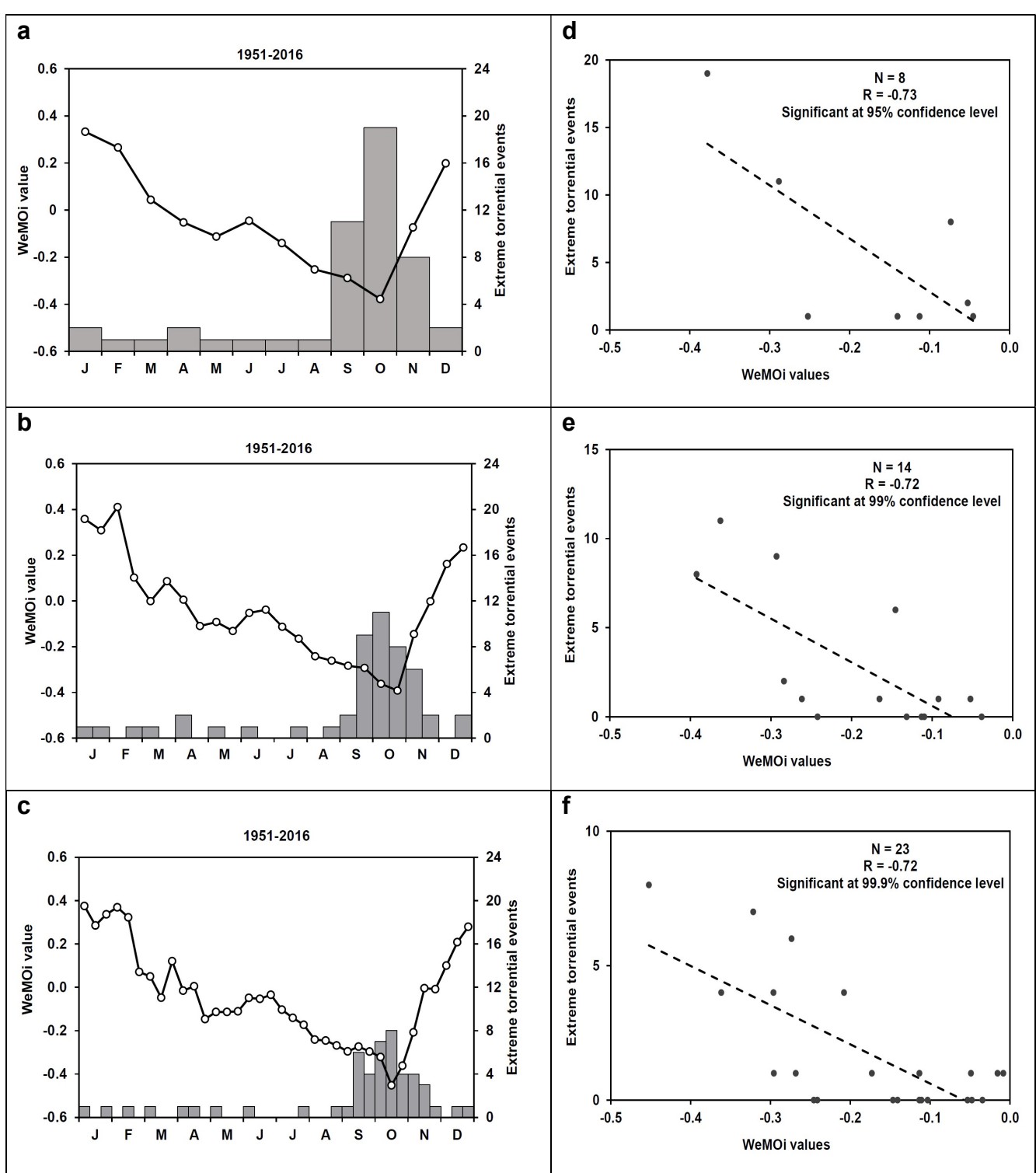

Figure 7. WeMOi calendars (lines) and frequency of extreme torrential episodes (bars) at several timescales: monthly (a), fortnightly (b) and 10-day (c). Scatterplot of the relationship between extreme torrential events and negative WeMOi values at several timescales: monthly (d), fortnightly (e) and 10-day (f); (the linear regression is shown as a dashed line).

(a)          (b)

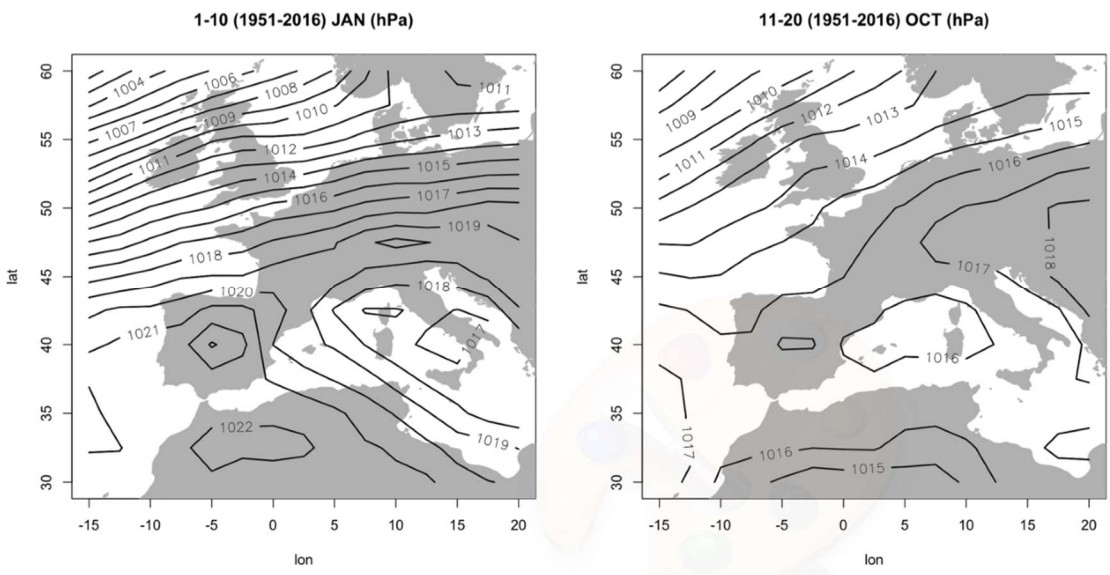


Figure 8. Sea level pressure (SLP) mean of the synoptic window 30ºN-60ºN and
15ºW-20ºE from January 1st to 10th (a) and from October 11th to 20th (b) during
the 1951-2016 study period. Data source: NCEP Reanalysis data provided by the
NOAA/OAR/ESRL PSD, Boulder, Colorado, USA.
The WeMO teleconnection pattern can exert its influence upon precipitation
variability in other regions of Southern Europe (Caloiero *et al*., 2011; Milosevic *et*
*al*., 2016; Mathbout *et al*., 2020). This central period of October may be the most
prone to torrential events over many regions of the western Mediterranean due
to presenting the lowest WeMOi value of the year. On the Iberian Peninsula, the
Almanzora river (SE Spain) suffered 2 of the 4 most catastrophic floods in the last
450 years within this central interval in October (Sánchez-García *et al*., 2019).
Moreover, the deadliest torrential episodes in the Valencia Region (E Spain)
occurred on October 13th-14th 1957 and October 19th-20th 1982 (Olcina *et al*.,
2016; Miró *et al*., 2017).

*4.3. Subperiods and differences in the calendars*
In relation to the calendars, and according to subperiods, we observed an overall
decrease in WeMOi values throughout the year (Figure 9). On the contrary, no
change was observed in the frequency of episodes between both subperiods;

exactly 25 extreme torrential events occurred in each subperiod. At the monthly timescale, the extreme torrential period takes place in September and October during the first half (1951-1983). For the second half (1984-2016), the maximum accumulation of cases shifts from September-October to October-November, with the highest concentration of cases in October, whilst new cases occur during early winter (December). All WeMOi values are statistically and significantly lower during the second subperiod than during the first one in all months, especially from October to December. In the summer months, the decrease in WeMOi values is moderate, albeit statistically significant due to the low variability of the WeMOi values during the warm months. All these seasonal changes can be related to trends in SST during the last few decades; the highest rate of SST warming is in November (0.42 °C per decade) (Table 2). Higher SST is directly associated both with a high rate of sea water evaporation and with more intense latent heat transfer to the atmosphere (Pastor *et al*., 2015), which is necessary with regard to greatly increasing the precipitable water in the column. A general warming of sea temperature has occurred along the year at all levels (SST, 20, 50, and 80 m.b.s.l.), particularly in spring, late autumn and early winter, a fact which might explain these more negative WeMOi values during the second subperiod; the warming of the lowest level of the atmosphere over the Western Mediterranean Sea contributes to the formation of mesoscale lows (Jansà *et al*., 2000). Similar rates of warming at near-surface sea level have been recorded in other locations in the north Mediterranean Sea (Raicich and Colucci, 2019). The highest warming rates have been observed at SST and 20 m.b.s.l., but the statistical significance has been greater at the deepest levels, i.e. 50 and 80 m.b.s.l. (Table 2). Figure 10 shows that changes in WeMOi values between both subperiods are negatively and statistically correlated with sea temperature trends, above all, in the underlying layers, especially at 80 m.b.s.l., where sea temperature displays a low interannual and intra-annual variability and sea heat content hardly varies (Sparnocchia *et al*., 2006).

At the fortnightly timescale, a shifting of maximum torrentiality is observed from September 16th – October 15th to October 1st – October 31st. The lowest WeMOi value of the calendar from 1951 to 1983 was in the first fortnight of October (-0.26); however, the lowest value is observed in the second fortnight of October

during the 1984-2016 period (-0.58). All WeMOi values according to fortnights showed a statistical and significant decrease during the second period, except from January 16th to March 15th. The sharpest decline in WeMOi values is in the first fortnight of May, the second fortnight of October, the second fortnight of November and the first fortnight of December. The lowest WeMOi value during the second subperiod is detected in the second fortnight of October, when the greatest increase in extreme torrential events is observed.

At the 10-day timescale the lowest WeMOi values remain relatively constant from the end of August to the beginning of November during the first subperiod, which corresponds well with the occurrence of extreme torrential events. During the second subperiod, the lowest WeMOi values are found from October 11th to 31st, with an accumulation of 8 cases (32% of the total number of cases of the second subperiod). A continuous and statistically significant decrease in WeMOi values (at the 99.9% confidence level) is observed from October 16th to December 20th during the second subperiod, except for the first 10-day period of November. The increase in torrential events is especially concentrated from October 21st to 31st. From August 21st to October 10th there is an overall decline in extreme torrential events, which might be associated with the fact that the WeMOi values hardly show a decrease over these 10-day periods of the year during the second subperiod. This is in line with the fact that the warming was moderate, or that there was even a certain degree of cooling, during the first 10-day periods of the wet season, i.e. from September 1st to October 20th, in the underlying sea layers (Table 3); and consequently, episodes might not have been favoured during the second subperiod. The highest sea temperature increase at all levels during the wet season is in the third 10-day period of October (Table 3), when the highest increase in extreme torrential episodes is observed (Figure 9). The changes in the frequency of episodes are statistically correlated with sea temperatures at subsurface layers, i.e. 50 and 80 m.b.s.l. (Figure 11). The deepest level (80 m.b.s.l.) shows the strongest warming in late autumn (from October 21st to November 30th), whereas this warming is weak in early autumn (from September 1st to October 20th) (Figure 12). This could be related to some recent changes in thermocline depth and time of destruction thereof due to warming of the Mediterranean Sea over the last few decades (Salat *et al.*, 2019). The subsurface

temperature may show a more constant warming of the Mediterranean Sea than
SST, because the latter is usually affected by local phenomena.
In general terms, no more cases of extreme torrential events are observed during
the 1984-2016 period in comparison with the 1951-1983 period. Nonetheless, a
greater accumulation of cases can be observed during late autumn and a lesser
accumulation in early autumn during the second subperiod, in comparison with
the first one. A sharp and continuous drop in WeMOi values is observed at the
very end of autumn, which might indicate a shift in the seasonality of the extreme
torrential period from September-October to October-November and an increase
in precipitation irregularity due to a deeper WeMO negative phase (Lopez-Bustins
and Lemus-Canovas, 2020). This seasonal shifting might be caused by a recent
increase in sea temperature in the Western Mediterranean basin, particularly in
November (Table 2) and late October (Table 3) (Lopez-Bustins, 2007; Estrela *et*
*al*., 2008; Lopez-Bustins *et al*., 2016; Arbiol-Roca *et al*., 2017). Pastor *et al*.
(2018) used satellite data to identify an overall increase in SST throughout the
Mediterranean basin during the 1982-2016 period, highlighting its role in torrential
events in the Western Mediterranean.








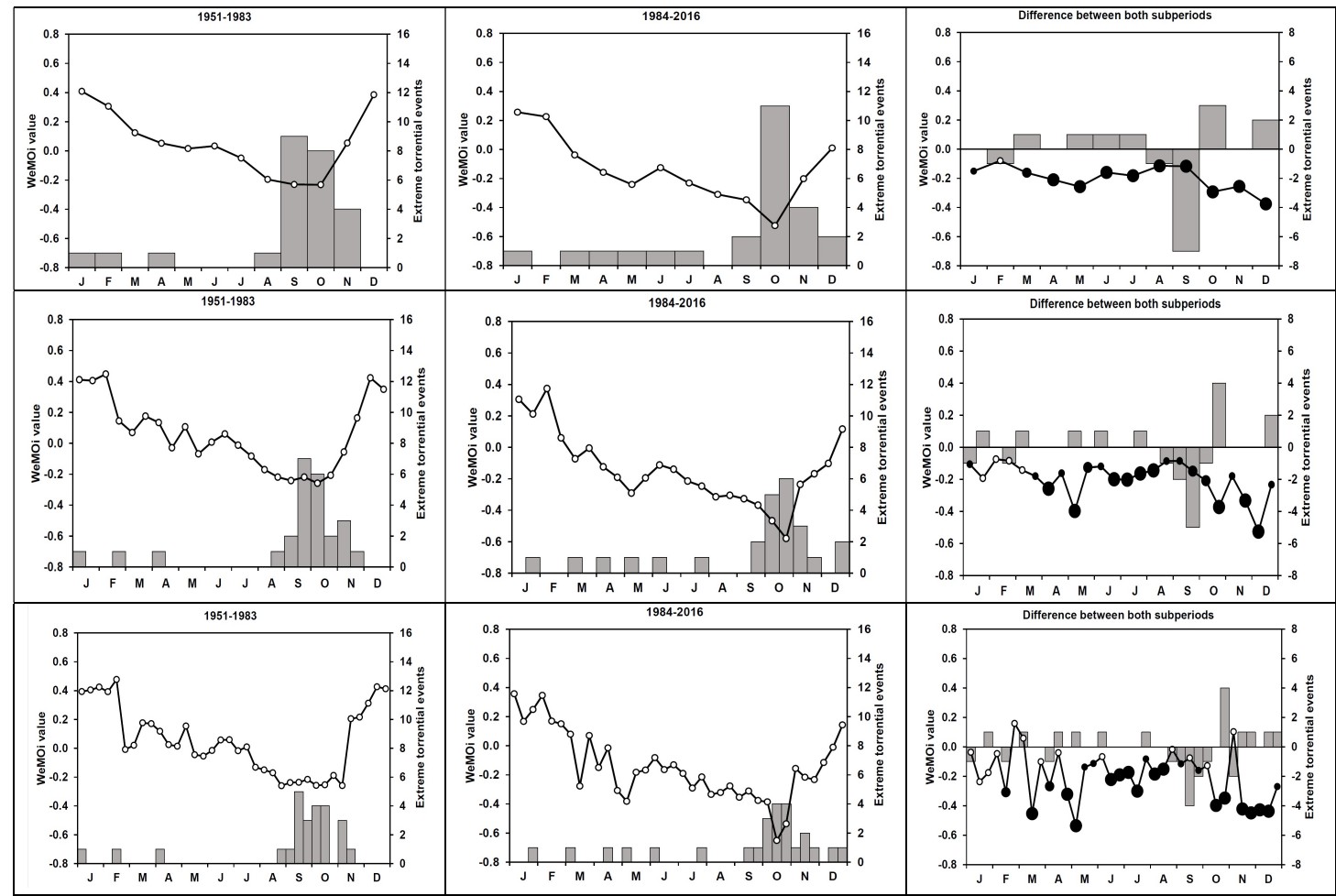

Figure 9. WeMOi calendars (lines) and frequency of extreme torrential episodes
(bars) at several timescales: monthly (above), fortnightly (middle) and 10-day
(below) for the 1951-1983 (left) and 1984-2016 (central) subperiods. The right-
hand column shows the difference in the number of episodes and WeMOi values
between both subperiods (for WeMOi values: white dots indicate not statistically
significant differences, and small-, medium- and large-sized black dots show
statistically significant differences at the 95.0%, 99.0% and 99.9% confidence
levels, respectively).





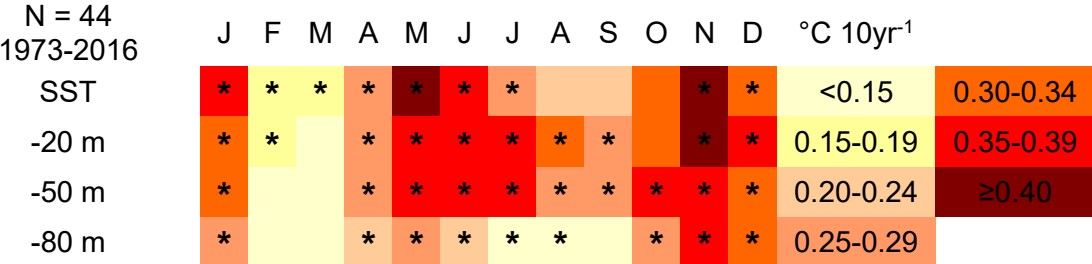

| N = 44 1973-2016 | J | F | M | A | M | J | J | A | S | O | N | D |
|---|---|---|---|---|---|---|---|---|---|---|---|---|
| SST | * | * | * | * | * | * | * |  |  |  | * | * |
| -20 m | * | * |  | * | * | * | * | * | * |  | * | * |
| -50 m | * |  |  | * | * | * | * | * | * | * | * | * |
| -80 m | * |  |  | * | * | * | * | * |  | * | * | * |

°C 10yr⁻¹ legend:

| °C 10yr$^{-1}$ | |
|---|---|
| <0.15 | 0.30-0.34 |
| 0.15-0.19 | 0.35-0.39 |
| 0.20-0.24 | ≥0.40 |
| 0.25-0.29 | |

Table 2. Monthly sea temperature trends at surface (SST), 20, 50, and 80 m.b.s.l. during 1973-2016 (*statistically significant trends at the 95% confidence level by means of the Mann-Kendall non-parametric test).

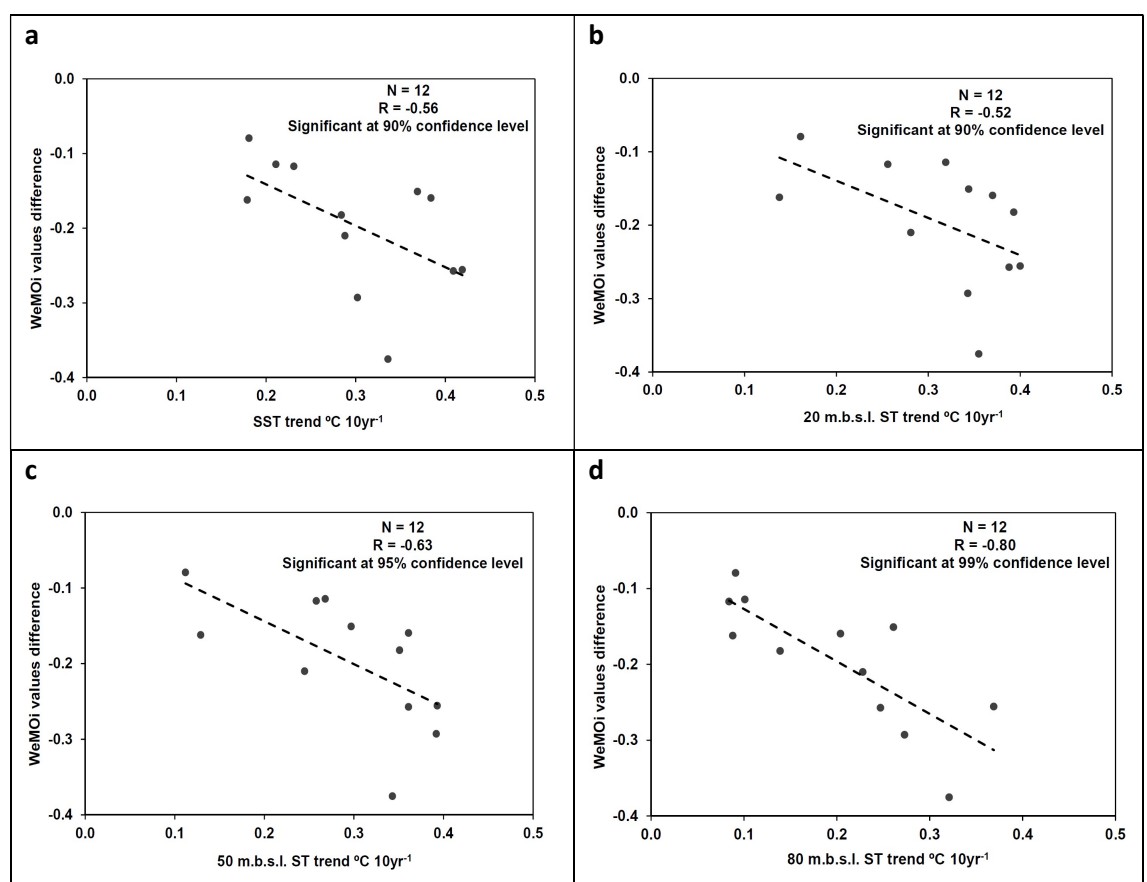

Figure 10. Scatterplot of the monthly relationship between the WeMOi value differences (1984-2016 minus 1951-1983) and sea temperature (ST) trends during the 1973-2016 period at surface (SST) (a), 20 (b), 50 (c), and 80 (d) m.b.s.l. (a dashed line indicates the linear regression).

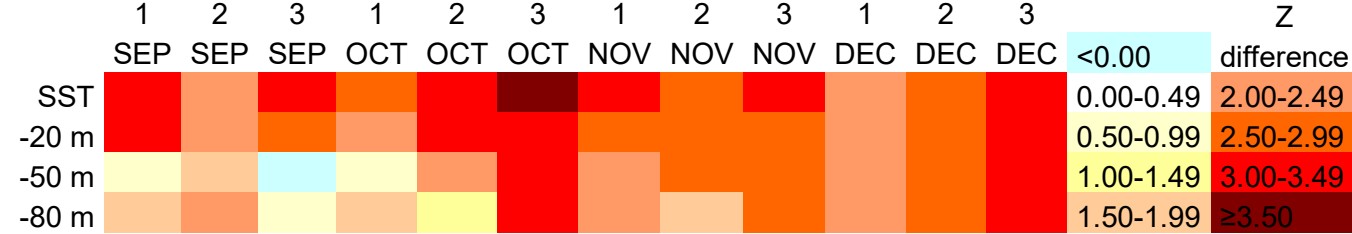

| | 1 SEP | 2 SEP | 3 SEP | 1 OCT | 2 OCT | 3 OCT | 1 NOV | 2 NOV | 3 NOV | 1 DEC | 2 DEC | 3 DEC | | Z difference |
|---|---|---|---|---|---|---|---|---|---|---|---|---|---|---|
| SST | | | | | | | | | | | | | <0.00 | 2.00-2.49 |
| -20 m | | | | | | | | | | | | | 0.00-0.49 | 2.50-2.99 |
| -50 m | | | | | | | | | | | | | 0.50-0.99 | 3.00-3.49 |
| -80 m | | | | | | | | | | | | | 1.00-1.49 | ≥3.50 |
| | | | | | | | | | | | | | 1.50-1.99 | |

Table 3. 10-day period ST standardized values (Z) differences for two 5-yr subperiods (2013-2017 minus 1973-1977) at surface, 20, 50, and 80 m.b.s.l. during the wet season (from September to November) and December.

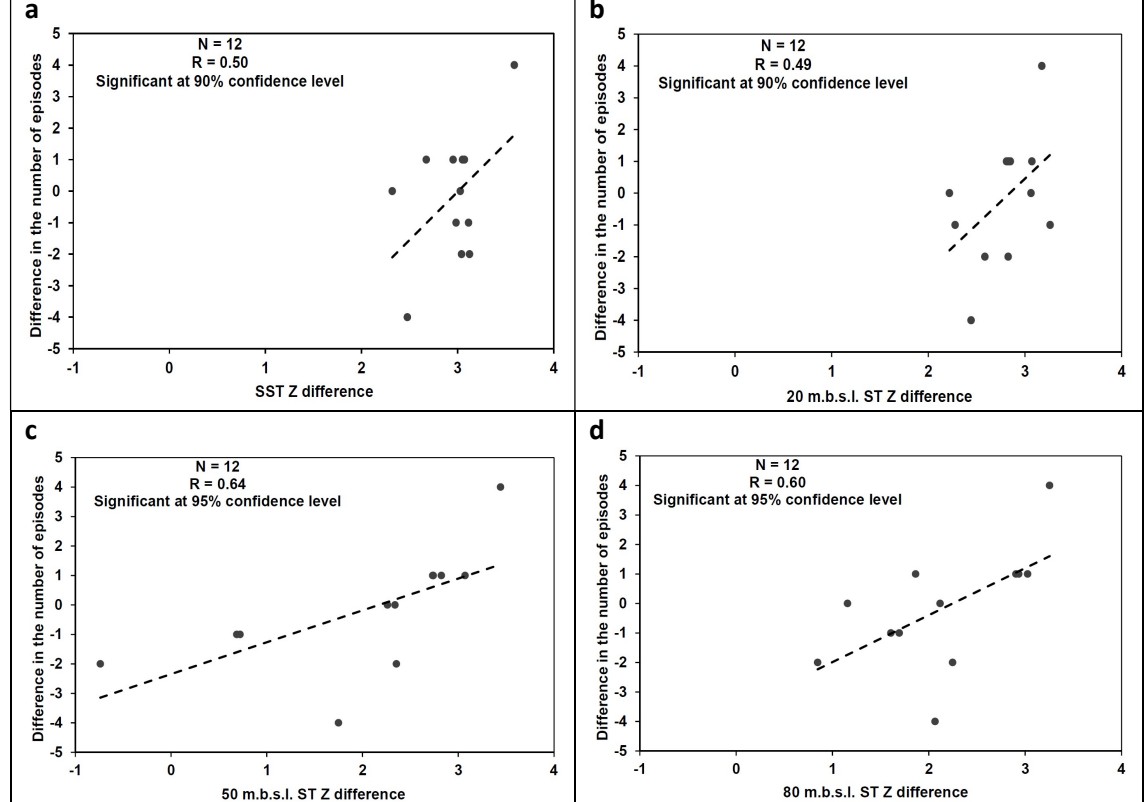

Figure 11. Scatterplot of the 10-day relationship between the differences in the number of episodes (1984-2016 minus 1951-1983) and ST Z differences for two 5-yr subperiods (2013-2017 minus 1973-1977) at surface (a), 20 (b), 50 (c), and 80 (d) m.b.s.l. during the wet season (from September to November) and December (a dashed line indicates the linear regression).

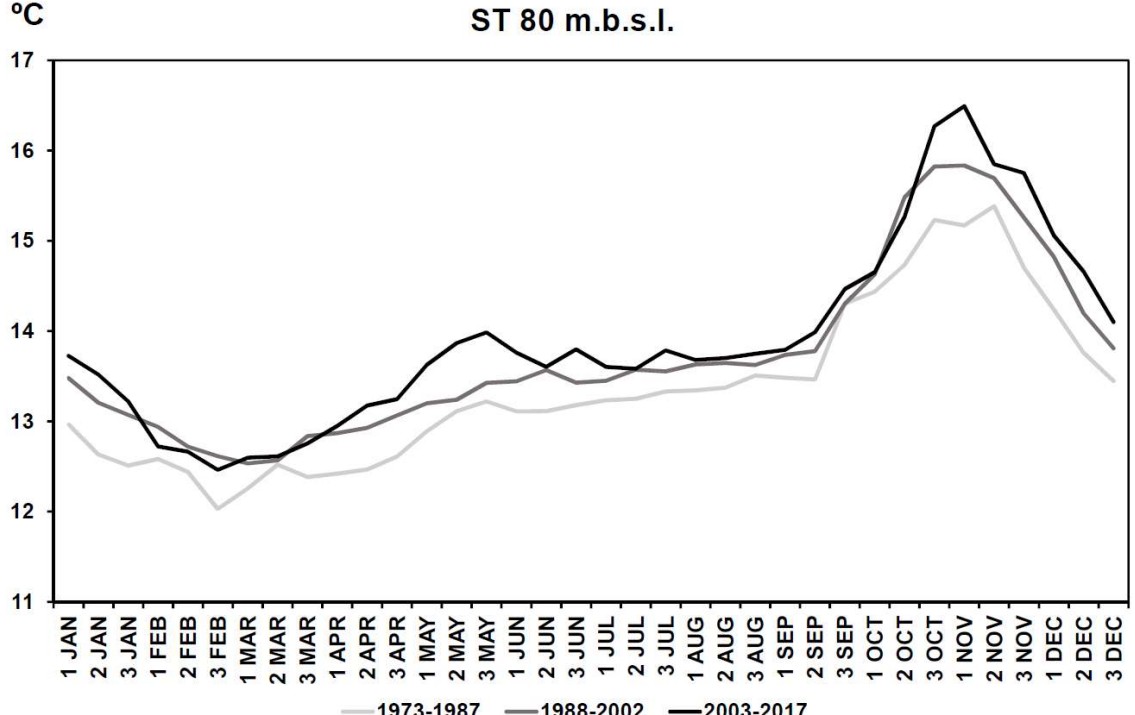


Figure 12. ST 10-day calendar at 80 m.b.s.l. for three 15-yr subperiods: 1973-
1987, 1988-2002 and 2003-2017.
**5. Conclusions**
The present research confirms the usefulness of the WeMOi at daily resolution
as an effective tool for analysing the occurrence of episodes of torrential
precipitation over NE Spain. October is the rainiest month in most regions of the
Northwestern Mediterranean basin and can account for the lowest value of the
year on the WeMOi monthly calendar, together with the warmest sea temperature
of the year at subsurface level. Moreover, most torrential episodes take place
during a very short period in the middle of this month.
Catalonia is located in the Northwestern Mediterranean basin and its extreme
precipitation is highly dependent upon the atmospheric circulation over the
Mediterranean. The present study considers the threshold of 200 mm in 24 h for
extreme torrential episodes, due to the fact that this precipitation accumulation in
one day can cause serious widespread damage over a large area. Having
thoroughly reviewed several databases and contrasted these results with the
original files and nearby weather stations, we confirmed that Catalonia registered

0.8 cases per year (50 episodes in 66 years) of extreme torrential episodes during the 1951-2016 study period, in accordance with the 7-7 UTC pluviometric day.

The 10-day period from October 11th to 20th exhibits both the greatest accumulation of extreme torrential episodes in Catalonia and the lowest intra-annual WeMOi value. This 10-day period has been demonstrated to be the most prone to torrential events in this Northwestern Mediterranean area, according to the WeMOi values. The most intense torrential event in Catalonia ever recorded by an official weather station is in Cape Creus (the easternmost part of the Iberian Peninsula) within the 10-day period most susceptible to torrential precipitation (October 13th 1986), with a total amount of 430 mm. The most positive WeMO phase of the year usually takes place in January, especially from January 1st to 10th, when the synoptic and sea temperature conditions of this time of the year inhibit torrential events.

No extreme torrential episodes in Catalonia occurred in a positive WeMO phase. Additionally, 60% of the cases occurred in an extreme negative WeMO phase, i.e. a WeMOi value equal to or lower than -2.00. In the present study this threshold is considered to constitute the onset of a rainstorm favoured by a strong Mediterranean flow. The lower WeMOi value is related to an increase in extreme torrential events at all timescales. On comparing both study subperiods (1951-1983 and 1984-2016), an overall statistically significant decrease is detected in most WeMOi values of the year, especially at the end of October and some periods in November and December. This might have been caused by an overall increase in sea temperature throughout the year, particularly in late autumn; this sea warming can enhance air convection (a decrease in surface pressure) over the Western Mediterranean basin. On the other hand, extreme torrential events show no changes in frequency between both subperiods; no temporal trend is observed, either, during the 1951-2016 study period. The most notable change involves the displacement of extreme torrential episodes from early to late autumn; this is in accordance with the lower WeMOi values detected in the last three months of the year during the second subperiod. Increases in sea temperatures in the underlying layers during the end of the wet season can provide an understanding of these changes in extreme torrential events and in the WeMOi calendars. Sea temperature is an additional factor influencing

torrential episodes in Catalonia; higher (lower) precipitation amounts can be registered in accordance with warmer (colder) than normal sea waters (Lebeaupin *et al*., 2006). The main causes of heavy precipitation in Catalonia involve easterly humid flows at surface level with an upper cut-off low (Martin-Vide *et al*., 2008), and troughs in the upper troposphere with an advection maximum of positive vorticity on their front edge (Lolis and Türkeş, 2016).

**Data availability**

The WeMOi data can be downloaded from the Climatology Group (University of Barcelona) website http://www.ub.edu/gc/en/ (last accessed July 5th 2020).

**Author contributions**

JALB performed the analysis and wrote the paper. LAR updated the WeMOi data and plotted the pressure maps. JMV discussed the results. ABE elaborated the inventory of the episodes and discussed the results. MPD discussed the results.

**Competing interests**

The authors declare that they have no conflict of interest.

**Acknowledgments**

The present study was conducted within the framework of the Climatology Group of the University of Barcelona (2017 SGR 1362, Catalan Government) and the CLICES Spanish project (CGL2017-83866-C3-2-R, AEI/FEDER, UE). Our research benefited from the daily precipitation data provided by the Meteorological Service of Catalonia. We are especially indebted to the meteorological observer from l'Estartit (Girona province), Josep Pascual, who painstakingly recorded sea temperature data over the last few decades.

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
