# Peer review of "Intra-annual variability of the Western Mediterranean Oscillation (WeMO)"

_Natural Hazards and Earth System Sciences, 2019_

## Referee Comment (RC1) · Joan Albert Lopez-Bustins et al. · 16 Dec 2019

Overview:

This manuscript addresses the occurrence of extreme torrential precipitation episodes in Catalonia (Northeast Spain). These episodes are considered as 24-hour periods with total precipitation amounts over 200 mm, rather than the commonly considered 100 mm threshold. The analysis is carried out from 1951 through 2016 (66 years) and using 70 weather stations covering Catalonia. A total of 50 episodes was identified and their occurrence was subsequently related to a teleconnection pattern index, the Western Mediterranean Oscillation index (WeMOi). These relationships are assessed

not only at the monthly timescale but also at two-week and 10-day timescales.

General comments:

The manuscript is clearly presented and the results are generally sounding and in line with previous studies. A satisfactory state-of-the-art is provided, giving credit to the most relevant preceding studies. Nonetheless, I found that the manuscript does not add significant new information to this topic of research. As it is currently, the manuscript is mostly a statistical description of the connections between extremes and WeMOi. From my viewpoint, the study lacks a more detailed analysis of the mechanisms underlying the occurrence of these events in Catalonia. The use of a single teleconnection index is too simplistic and does not bring any added value to both the forecast of these events and to their understanding. More focus should be given to mesoscale processes and dynamical features, also highlighting singularities.

Specific comments:

1. I recommend replacing "rainfall" with"precipitation" throughout the text, as e.g. hailfall may have occurred on some occasions.

2. Keywords are too vague. Please revise.

3. Lines 93-101: The authors state that: "The main aim of the study involves establishing a period of high potential torrentiality in Catalonia at daily resolution" and below that "Therefore, the present research attempts to go beyond the monthly timescale in order to determine the period with the highest accumulation of heavy rainfall according to fortnights 99 and 10-day periods. The intra-annual variability of the daily WeMOi values may help to establish the period with the highest propensity for torrential events in Catalonia". As previously mentioned, from my point of view this single objective of the study is not enough to justify the publication of the study. A much more detailed analysis should be provided, including an analysis of dynamical precursors, which would be very important for improving weather forecasts and the general understanding of these

events.

4. Ln 108: the authors mention several times "south of France", but the weather stations located in France only cover a very limited area of southern France. Hence, this terminology is a bit misleading and should be revised. Furthermore, the analysis for the French stations does not bring any significant new information and should be discarded from the study. Further, a different threshold is used (100 mm), as is said in Ln 471, thus not allowing a comparison.

5. Fig. 3: The use of NCEP reanalysis is not the best option. The ERA5 dataset should be used instead. Also, the quality of the panels should be considerably improved.

6. Ln 368: five consecutive days? Fig. 6 shows 5 instead of 4. Please clarify.

7. Fig. 7 and subsequent: the means of the bars and lines are not explained in the panels. Please revise.

8. The 2-order polynomial fitting is not duly explained. What is the purpose of these adjustments? What can be concluded from them?

9. Ln 584-586: The authors mention that "Further research on this theme 584 is required and SST temporal trends might provide a better understanding of these changes in extreme torrential events and WeMOi calendars". This type of analysis should not be left to a forthcoming study. This is a good suggestion to improve the manuscript.

Technical comments:

1. Please replace "furnished" by "provided" or similar throughout the text.

2. The overall quality and resolution of the figures should be improved.

---

## Referee Comment (RC2) · Anonymous Referee #2 · 8 Jan 2020

General Overview:

The authors analyzed the intra-annual variability of the Western Mediterranean Oscillation and occurrence of extreme torrential rainfall in Catalonia (NE Iberia). Despite the target region and topic is of interest to be study due the possible socio-economic impacts of the torrential rainfall, the manuscript in the present form do not add much to the present knowledge. In addition, it has some very important methodological and organizational issues which are listed below:

1) My main concern is that the manuscript fails to add new knowledge to the literature. In the present form, the manuscript is rather descriptive specially in section 4.2 and 4.3 where there is a statistical description between WeMO and the torrential rain which was previously known. From my point of view, there is the lack of understanding what is the physical mechanism which are behind the extreme torrential rainfall in Catalonia, for example, the atmospheric forcing, the role of SST, or even the soil moisture availability.

2) Figure 2a) is computed with data from where? The monthly series provided by the Meteorological Service of Catalonia?

3) The authors use a fix threshold to define the extreme torrential episodes which is >200mm in 24h. L168-173. I do not agree with this sentence. Based on my experience I can imagine that precipitation >100mm in a relative larger area will have more impacts than a precipitation >200mm only recorded in one single weather station. Therefore, I encourage the authors to think of a way to define the torrential episodes based not only on the amount of precipitation but also on it's spatial extent.

4) There is an inconsistent between the period of analyses. On line L126 is mentioned 1950-2015 and on L167 1950-2016.

5) The authors need to include a better description of the weather stations. How many of them are at a daily scale vs semi-hourly data. Since which year do you have access to automatic weather stations?

6) L220-222 The WeMo is computed using SLP from the weather stations mentioned in the text? They are quality controlled?

7) In Figure 3 and Figure 8 the authors used the outdated NCEP/NCAR reanalysis. Please use ERA5 instead.

8) Figure 7 d , e ,f ). These results are not mentioned in the text. I would exclude it from the manuscript.

9) L268 The mean and standard deviation is computed at an annual scale or at a day

level?

10) Figure 4. Why this division?

11) Regarding section 3.3, why don't the authors use a moving average instead of artificial 10-day or 15-days intervals?

12) L468-470. I don't think that 4 weather stations are representative of southern France. I would delete everything related with these 4 weather stations from the text, including Figure 9.

13) L527-529 I agree with the authors and I think an analysis on this, among physical mechanisms (see comment 1), should be included in the new version of the manuscript.

Therefore, I recommend the major revision of the manuscript.

––––––––––––––––––––––––––––––––

---

## Author Comment (AC1) · 31 May 2020

Anonymous Referee #1 "Author's response"

Overview:

This manuscript addresses the occurrence of extreme torrential precipitation episodes in Catalonia (Northeast Spain). These episodes are considered as 24-hour periods with total precipitation amounts over 200 mm, rather than the commonly considered 100 mm threshold. The analysis is carried out from 1951 through 2016 (66 years) and

using 70 weather stations covering Catalonia. A total of 50 episodes was identified and their occurrence was subsequently related to a teleconnection pattern index, the Western Mediterranean Oscillation index (WeMOi). These relationships are assessed not only at the monthly timescale but also at two-week and 10-day timescales.

"We are very grateful for the reviewer's comments and for the revision of our manuscript. The paper has been revised in accordance with the referee's comments and suggestions, which are addressed below. Our answers appear within quotation marks".

General comments:

The manuscript is clearly presented and the results are generally sounding and in line with previous studies. A satisfactory state-of-the-art is provided, giving credit to the most relevant preceding studies. Nonetheless, I found that the manuscript does not add significant new information to this topic of research. As it is currently, the manuscript is mostly a statistical description of the connections between extremes and WeMOi. From my viewpoint, the study lacks a more detailed analysis of the mechanisms underlying the occurrence of these events in Catalonia. The use of a single teleconnection index is too simplistic and does not bring any added value to both the forecast of these events and to their understanding. More focus should be given to mesoscale processes and dynamical features, also highlighting singularities.

"The main contribution of the paper involves an accurate database of extreme torrential episodes. It was a painstaking task to select the appropriate episodes, as well as to review several databases and handwritten cards. Indeed, we consider this to constitute the most reliable extreme torrential database existing for this region".

"We agree with the reviewer in that dynamical mechanisms are lacking; we have therefore included new analyses that consider the temporal evolution of sea temperature from one specific high-quality series on the coast covering several decades (1973-2017) (please see tables 2 and 3, and figures 10, 11 and 12). The results show a

statistical relationship between changes in the WeMOi and SST trends. Furthermore, we have added a long text explaining the dynamical mechanisms in the introduction on L88-114 and we have included many new references. We have also added three references from 2020".

"We agree with the reviewer that it is too simplistic to use only one teleconnection index. We have added NAOi values for figures 4 and 5 in order to demonstrate the better fit of the WeMOi in comparison to the NAOi, and these are commented on in the text on L323-331 and L404-411".

Specific comments:

1. I recommend replacing "rainfall" with "precipitation" throughout the text, as e.g. hailfall may have occurred on some occasions.

"We agree with the reviewer. We have replaced it throughout the manuscript".

2. Keywords are too vague. Please revise.

"We have changed some of them. The current keywords are Mediterranean, sea temperature, teleconnection indices, torrential precipitation, WeMO".

3. Lines 93-101: The authors state that: "The main aim of the study involves establishing a period of high potential torrentiality in Catalonia at daily resolution" and below that "Therefore, the present research attempts to go beyond the monthly timescale in order to determine the period with the highest accumulation of heavy rainfall according to fortnights 99 and 10-day periods. The intra-annual variability of the daily WeMOi values may help to establish the period with the highest propensity for torrential events in Catalonia". As previously mentioned, from my point of view this single objective of the study is not enough to justify the publication of the study. A much more detailed analysis should be provided, including an analysis of dynamical precursors, which would be very important for improving weather forecasts and the general understanding of these events.

"We have rewritten the main aim of the study on L123-125 to highlight the importance of the creation of the catalogue. Furthermore, new analyses involving SST have been added (Tables 2 and 3, and Figures 10, 11 and 12)".

4. Ln 108: the authors mention several times "south of France", but the weather stations located in France only cover a very limited area of southern France. Hence, this terminology is a bit misleading and should be revised. Furthermore, the analysis for the French stations does not bring any significant new information and should be discarded from the study. Further, a different threshold is used (100 mm), as is said in Ln 471, thus not allowing a comparison.

"We agree with the reviewer and we have discarded it from the study".

5. Fig. 3: The use of NCEP reanalysis is not the best option. The ERA5 dataset should be used instead. Also, the quality of the panels should be considerably improved.

"ERA5 is a better (higher resolution and a more complete global circulation model), updated reanalysis in comparison with the NCEP/NCAR reanalysis, but ERA5 currently only covers the time period from 1979. Therefore, we are unable to redesign figure 3 (a) and figures 8 (a) and 8 (b). Moreover, the definition of spatial resolution is not relevant with regard to shaping the WeMO phase occurring on these days. Nonetheless, we have improved the quality of all figures with NCEP/NCAR reanalysis".

6. Ln 368: five consecutive days? Fig. 6 shows 5 instead of 4. Please clarify.

"For clarity, we have modified the sentence as follows "The greatest accumulation of cases can be observed in 1971, when a long-lasting torrential episode exceeded the threshold of 200 mm in 24 h during four consecutive days in September, with another one-day episode occurring in October" (L417-420)".

7. Fig. 7 and subsequent: the means of the bars and lines are not explained in the panels. Please revise.

"To clarify it we have modified the sentence in the caption of the figures 7 and 10

as follows "WeMOi calendars (lines) and frequency of the extreme torrential episodes (bars) at several timescales"".

8. The 2-order polynomial fitting is not duly explained. What is the purpose of these adjustments? What can be concluded from them?

"We have checked why we used the 2nd-order polynomial fitting. We did so following a simple visual inspection, but it makes little physical sense. There is no atmospheric reason for an increase in extreme torrential events with positive WeMOi values. We have therefore calculated the regression line for only the WeMOi negative values, after verifying the statistically significant correlation between episodes and the WeMOi. In Figure 7 (d, e, and f) we have replaced the quadratic fit with the linear fit, and accordingly, we have done the same in the caption of the figure and in the text L497-504. The linear fit is especially significant at 10-day resolution. There is an evident increase in the occurrence of events with a decrease in WeMOi values".

9. Ln 584-586: The authors mention that "Further research on this theme 584 is required and SST temporal trends might provide a better understanding of these changes in extreme torrential events and WeMOi calendars". This type of analysis should not be left to a forthcoming study. This is a good suggestion to improve the manuscript.

"We have included new analyses considering the temporal evolution of SST from one specific high-quality station on the coast which covers several decades (1973-2017) (please see Tables 2 and 3, and Figures 10, 11 and 12)".

Technical comments:

1. Please replace "furnished" by "provided" or similar throughout the text.

"Done".

2. The overall quality and resolution of the figures should be improved.

"To this end we have redesigned all the figures".

---

## Author Comment (AC2) · 31 May 2020

Anonymous Referee #2 "Author's response"

General Overview:

The authors analyzed the intra-annual variability of the Western Mediterranean Oscillation and occurrence of extreme torrential rainfall in Catalonia (NE Iberia). Despite the target region and topic is of interest to be study due the possible socio-economic impacts of the torrential rainfall, the manuscript in the present form do not add much

to the present knowledge. In addition, it has some very important methodological and organizational issues which are listed below:

"We wish to thank the reviewer for his/her comments and for reviewing our manuscript. The manuscript has been revised in consonance with the referee's comments and suggestions, which are addressed below. Our answers appear within quotation marks".

1) My main concern is that the manuscript fails to add new knowledge to the literature. In the present form, the manuscript is rather descriptive specially in section 4.2 and 4.3 where there is a statistical description between WeMO and the torrential rain which was previously known. From my point of view, there is the lack of understanding what is the physical mechanism which are behind the extreme torrential rainfall in Catalonia, for example, the atmospheric forcing, the role of SST, or even the soil moisture availability.

"We agree with the reviewer regarding the lack of physical mechanisms; consequently, we have included new analyses, considering the temporal evolution of sea temperature from one specific high-quality series on the coast which encompasses several decades (1973-2017) (please see tables 2 and 3, and figures 10, 11 and 12). The results show a statistical relationship between changes in the WeMOi and SST trends. Furthermore, we have added a text explaining the atmospheric mechanisms related to mesoscale convective systems in the Western Mediterranean, which justifies the application of WeMO calendars (L88-114). Moreover, we have added new references to the introduction. We have also added three references from 2020. Furthermore, we have added NAOi values for figures 4 and 5 in order to demonstrate the better fit of the WeMOi in relation to that of the NAOi".

2) Figure 2a) is computed with data from where? The monthly series provided by the Meteorological Service of Catalonia?

"Figure 2a is extracted from the 70 precipitation monthly series computed by the SMC (Yearly Bulletin of Climate Indicators, 2017) which have been quality controlled and analysed for homogeneity. The caption of Figure 2 now includes a paragraph explaining

the origin of the data. The study period has been changed from 1950-2015 to 1951-2016. The caption now reads "Figure 2. (a) Annual mean precipitation (mm) and (b) seasonal precipitation regimes for 70 weather stations in Catalonia for the 1951-2016 study period. Data source: SMC (2017). Base map provided by the Cartographic and Geological Institute of Catalonia"".

3) The authors use a fix threshold to define the extreme torrential episodes which is >200mm in 24h. L168-173. I do not agree with this sentence. Based on my experience I can imagine that precipitation >100mm in a relative larger area will have more impacts than a precipitation >200mm only recorded in one single weather station. Therefore, I encourage the authors to think of a way to define the torrential episodes based not only on the amount of precipitation but also on its spatial extent.

"Thank you for your comment, which has brought us to further reflect upon the thresholds defining torrential rainfall in the Mediterranean. We partly agree with your comment, but episodes presenting precipitation ≥100 mm/day in a relatively larger area are not so common in Catalonia, and when they do occur, they do not cause major damage or destruction. For example, Gilabert and Llasat (2018), one of the new references we have included, found that catastrophic flood events (rivers overflowing with major damage or total destruction), associated with extreme torrential precipitation events, are generally of synoptic origin and are enhanced by certain mesoscale factors, a phenomenon that is clearly reflected by the negative phase of the WeMO. We have chosen the threshold of ≥200 mm in one single weather station as a maximum value in order to capture the most important torrential precipitation events, but within these, the area affected by precipitation values ≥100 mm is sizeable. This area usually encompasses a significant part of Catalonia (almost one third). Further information in this respect has been added to the new manuscript on L109-114, L193-201 and L659-661".

"Moreover, in the first paragraph of subsection 3.1. 'Selection of the torrential events', we have further distinguished between 'torrential events' (threshold of ≥100 mm/24 h), widely used by Spanish authors, and 'extreme torrential events' (≥200 mm/24 h),

already used in several previous studies, particularly in Lopez-Bustins et al (2016), cited in the References and also in Martin-Vide (2002), one of the new references we have included, as well as in others, with good results. It is true that the spatial domain of heavy precipitation conditions the fluvial response and the possibility of flooding, and the combination of precipitation amounts and area affected therefore enables a more complete hydrological analysis than when only precipitation amounts are used. In the future, the authors may intend to explore a hydrological definition of torrential precipitation for the western Mediterranean basin, taking into account both precipitation values and area affected".

4) There is an inconsistent between the period of analyses. On line L126 is mentioned 1950-2015 and on L167 1950-2016.

"Many thanks for the observation. Indeed, there is an inconsistency that has been rectified both in the text (L162) and in Figure caption 2. The correct study period is 1951-2016".

5) The authors need to include a better description of the weather stations. How many of them are at a daily scale vs semi-hourly data. Since which year do you have access to automatic weather stations?

"There were 749 weather stations at daily timescale (manual) and 305 at hourly or semi-hourly timescale (automatic) throughout Catalonia during the 1951-2016 period. The 1951-1987 period was covered by manual weather stations only. The 1988-2016 period was covered by both manual and automatic weather stations. We specified this information on L205-211".

6) L220-222 The WeMo is computed using SLP from the weather stations mentioned in the text? They are quality controlled?

"Yes, it is (L243-247). Yes, they are (we have added new text to specify this L247-250)".

7) In Figure 3 and Figure 8 the authors used the outdated NCEP/NCAR reanalysis.

Please use ERA5 instead.

"ERA5 is a better (higher resolution and a more complete global circulation model), updated reanalysis in comparison with the NCEP/NCAR reanalysis, but ERA5 currently only covers the time period from 1979. Therefore, we are unable to redesign figure 3 (a) and figures 8 (a) and 8 (b). Moreover, the definition of spatial resolution is not relevant with regard to shaping the WeMO phase occurring on these days. Nonetheless, we have improved the quality of all figures with NCEP/NCAR reanalysis".

8) Figure 7 d , e ,f ). These results are not mentioned in the text. I would exclude it from the manuscript.

"Following the suggestion of the other reviewer, we have checked why we used the 2nd-order polynomial fit. We did so after a simple visual inspection, but it makes little physical sense. There is no atmospheric reason for an increase in extreme torrential events presenting positive WeMOi values. We have therefore calculated the regression line for only the WeMOi negative values, having verified the statistically significant correlation between episodes and the WeMOi. In Figure 7 (d, e, and f) we replaced the quadratic fit with the linear fit, and accordingly, we did the same in the figure caption; the figures are now commented in the text on L497-504. The linear fit is especially significant at a 10-day resolution. There is an evident increase in the occurrence of events with a decrease in WeMOi values".

9) L268 The mean and standard deviation is computed at an annual scale or at a day level?

"They are computed at a day level. We have included "daily" on L286".

10) Figure 4. Why this division?

"Because we have already used it in previous studies and the results were sound (Martin-Vide and Lopez-Bustins, 2006; Azorin-Molina and Lopez-Bustins, 2008). These references are included in the bibliography and in the manuscript L301-302".

11) Regarding section 3.3, why don't the authors use a moving average instead of artificial 10-day or 15-days intervals?

"In the present paper we used moving averages to perform an inter-annual analysis of the frequency of extreme torrential events (Figure 6). The construction of a calendar involved an intra-annual analysis based on climatological means. In this case, in addition to the simple monthly frequency, we preferred to use the half-monthly and the 10-day frequencies. The relative scarcity and temporal randomness of extreme torrential events at daily resolution reveal many "saw teeth", which are of no climatic or statistical significance".

12) L468-470. I don't think that 4 weather stations are representative of southern France. I would delete everything related with these 4 weather stations from the text, including Figure 9.

"We agree with the reviewer and we have discarded it from the study".

13) L527-529 I agree with the authors and I think an analysis on this, among physical mechanisms (see comment 1), should be included in the new version of the manuscript.

"We have included new analyses considering the temporal evolution of SST from one specific high-quality station on the coast covering several decades (1973-2017) (please see Tables 2 and 3, and Figures 10, 11 and 12)".

Therefore, I recommend the major revision of the manuscript

---

## Author Response (AR2)

We are very grateful for the comments by reviewer 2. The paper has been revised in accordance with the referee's comments and suggestions, which are addressed below. Our answers appear in bold.

1) Despite the inclusion of the SST analysis, I believe that the authors should comment what other physical mechanisms can trigger the extreme rainfall. A paragraph in the conclusions discussing this with the supporting references would improve the conclusion section.

**A paragraph has been added at the end of the conclusions section (L720-726). A new reference has been included in the article: (Lolis and Türkeş, 2016).**

2) Figure 9 needs to be improved. The readability of the X axis is very poor.

**The X axes of figure 9 (middle and below) have been redesigned. We have done the same thing with the figures 7b and 7c.**

3) Can the author comment why the use of SST and subsurface temperature at several depths (20, 50, and 80 m.b.s.l.) ??

**Because this is a reference series due its length and to the fact that it provides data at several depth levels. According to a recent NASA-funded study, this series constitutes a notable exception worldwide, as well as a reliable source for satellite data validation. We considered this to represent a good opportunity to study the role played by sea temperature at subsurface level in the occurrence of extreme precipitation in Catalonia. We highlighted the relevance of this series on L360-362 and the reference of the NASA-funded study: Salat *et al*. (2019). See the following NASA post: https://climate.nasa.gov/blog/2997/sea-change-why-long-records-of-coastal-climate-matter/**

4) Can the author comment why this occur? "The changes in the frequency of episodes are statistically correlated with sea temperatures at subsurface layers, i.e. 50 and 80 m.b.s.l. (Figure 11). The deepest level (80 m.b.s.l.) shows the strongest warming in late autumn (from October 21st to November 30th)".

**From the text and analysis of the new reference -Salat *et al*. (2019)-, we inferred that this is related to recent changes in the thermocline. Subsurface sea temperature may be playing an important role in the warming patterns of the Mediterranean Sea. This is explained in the new text on L606-610.**

5) There is the lack of physical interpretation regarding the SST statistical results. Can the authors improve this particular part of the text?

**We added some information to further explain this on L554-557 and L710-712.**